# FROM LINK PREDICTION TO FORECASTING: INFORMATION LOSS IN BATCH-BASED TEMPORAL GRAPH LEARNING

## ABSTRACT

Dynamic link prediction is an important problem considered by many recent works proposing various approaches for learning temporal edge patterns. To assess their efficacy, models are evaluated on publicly available benchmark datasets involving continuous-time and discrete-time temporal graphs. However, as we show in this work, the suitability of common batch-oriented evaluation depends on the datasets' characteristics, which can cause multiple issues: For continuous-time temporal graphs, fixed-size batches create time windows with different durations, resulting in an inconsistent dynamic link prediction task. For discrete-time temporal graphs, the sequence of batches can additionally introduce temporal dependencies that are not present in the data. In this work, we empirically show that this common evaluation approach leads to skewed model performance and hinders the fair comparison of methods. We mitigate this problem by reformulating dynamic link prediction as a *link forecasting* task that better accounts for temporal information present in the data. We provide implementations of our new evaluation method for commonly used graph learning frameworks.

## 1 INTRODUCTION

Many scientific fields study data that can be modeled as graphs, where nodes represent entities that are connected by edges. Examples include social (Lazer et al., 2009), financial (Bardoscia et al., 2021), biological (Davidson et al., 2002) as well as molecular networks (David et al., 2020). Apart from the mere topology of interactions, i.e., who is connected to whom, such data increasingly include information on *when* these interactions occur. Depending on the temporal resolution, the resulting *temporal graphs* are often categorized as *continuous-time* or *discrete-time* (Longa et al., 2023): State-of-the-art data collection technology provides high-resolution *continuous-time* temporal graphs, which capture the exact (and possibly unique) occurrence time of each interaction. Examples include time-stamped online interactions (Kumar et al., 2019) or social networks captured via high-resolution proximity sensing technologies (Vanhems et al., 2013). In contrast, discrete-time temporal graphs give rise to a temporally ordered sequence of *static* snapshots, where each snapshot contains interactions recorded within a (typically coarse-grained) time interval. Examples include scholarly collaboration or citation graphs, which frequently include monthly or yearly snapshots.

Building on the growing importance of temporal data and the success of graph neural networks (GNNs) for static graphs (Bronstein et al., 2017; Corso et al., 2024), deep graph learning has recently been extended to temporal (or dynamic) graphs (Feng et al., 2024). To this end, several temporal graph neural network (TGNN) architectures have been proposed that are able to simultaneously learn temporal and topological patterns. These architectures are often evaluated in *dynamic link prediction*, where the task is to predict the existence of edges during a future time window of length $\Delta t$, e.g., to provide recommendations to users (Kumar et al., 2019).

For dynamic link prediction, TGNNs commonly utilize *temporal batches* to speed up training (Su et al., 2024). To construct these temporal batches, the sequence of *temporally* ordered edges is divided into a sequence of equally large chunks that contain the same number of edges. Within each batch, edges are typically treated as if they occurred simultaneously, thus discarding temporal information within a batch. For continuous-time temporal graphs, such fixed-size batches are likely associated

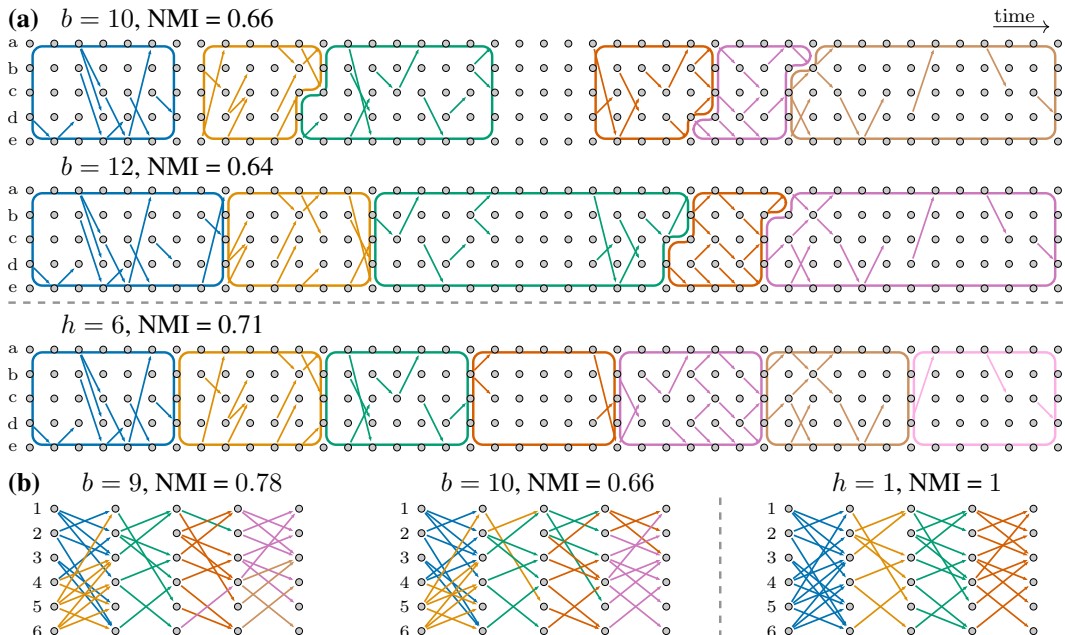

Figure 1: Illustration of the issues with a batch-based evaluation of TGNNs: **(a)** A continuous-time temporal graph, split into batches with sizes $b = 10$ (top), $b = 12$ (middle), and time windows with duration $h = 6$ (bottom). **(b)** A discrete-time temporal graph, split into batches with size $b = 9$ (left), $b = 10$ (middle), and time windows with duration $h = 1$ (right). Splitting temporal graphs with inhomogeneous temporal activities into batches with fixed size $b$ assigns edges in time windows of varying lengths to the same batch and edges with identical timestamps to different batches. We use normalized mutual information (NMI) between the edges' timestamps and their associated batch number (shown by colors) to quantify how much temporal information can be recovered from the sequence of batch numbers alone. In our work, we propose a time-window-oriented approach to evaluate dynamic link prediction that mitigates the information loss of current batch-based evaluation.

with time windows of varying lengths $\Delta t_i \neq \Delta t_j$. Changing the batch size affects the resulting window lengths and could, e.g., change the task from predicting at the minute to the hour level, thus altering its difficulty (see Figure 1a). In discrete-time temporal graphs, snapshots are typically so large that they comprise multiple batches (see Figure 1b). Thus, the ordered sequence of batches does not necessarily correspond to a *temporally ordered* sequence. In essence, batch-wise training of TGNNs effectively mixes information from the past, present, and future. This violates the arrow of time and questions the applicability of TGNNs in real-world prediction settings, where models do not have access to future information.

Addressing these important problems in the evaluation of temporal graph learning techniques, our work makes the following contributions:

- We quantify the information loss due to the aggregation of edges into batches on 14 discrete-time and continuous-time temporal graphs, thus showing how the dynamic link prediction task depends on the batch size.

- To better account for the arrow of time in the evaluated datasets, we formulate the task as *link forecasting* using a time-window-oriented evaluation that adequately considers the available temporal information that replaces the current link prediction task.

- We perform an experimental evaluation of state-of-the-art TGNNs for *link forecasting*. Our results highlight substantial differences in model performance compared to a batch-oriented evaluation of link prediction, thus demonstrating the real-world impact of our work. Furthermore, our results suggest that memory-based methods are not well suited for discrete-time data, which has so far been overlooked due to the overestimation of model performance caused by information leakage in batch-based evaluation.

While batch-oriented processing is a technical necessity for efficient model training, our work shows that tuning the batch size essentially tunes the link prediction task, thus fitting the task to the model and undermining a fair comparison of temporal graph learning techniques. Proposing a time-window-oriented evaluation of dynamic link *forecasting*, our work provides a simple yet effective solution, facilitating a fairer and more realistic evaluation approach that better reflects real-world scenarios.

## 2  PRELIMINARIES AND RELATED WORK

**Temporal graphs.**  A temporal (or dynamic) graph $G = (V, E)$ is a tuple where $V$ is the set of $n = |V|$ nodes and $E$ is a chronologically ordered sequence of $m = |E|$ time-stamped edges defined as $E = \{(u_0, v_0, t_0), \ldots, (u_{m-1}, v_{m-1}, t_{m-1})\}$ with $1 \leq t_0 \leq \cdots \leq t_{m-1} \leq t_{\max}$ (Poursafaei et al., 2022; Wang et al., 2021b; Yu et al., 2023). Each node $v_i$ can have static node features $\mathbf{h_i} \in H_V$ and each edge $(u_i, v_j, t)$ can have edge features $\mathbf{e_{ij,t}} \in H_E$ that change over time. We assume that interactions occur instantaneously with discrete timestamps $t \in \mathbb{N}$. Although timestamps $t \in \mathbb{N}$ are discrete, such temporal graphs are often categorized as continuous-time (Kazemi et al., 2020; Skarding et al., 2021). In contrast, discrete-time temporal graphs coarse-grain time-stamped edges into a sequence of static snapshot graphs $\{G_{t_i:t_j}\}$, where $G_{t_i:t_j} = (V, E_{t_i:t_j})$ with $E_{t_i:t_j} = \{(u, v) \mid \exists (u, v, t) \in E : t_i \leq t \leq t_j\}$ (Xue et al., 2022).

**Dynamic link prediction.**  Given time-stamped edges up to time $t$, the goal of dynamic link prediction is to predict whether an edge $(v, u, t + 1)$ exists at future time $t + 1$ (Yu et al., 2023; Poursafaei et al., 2022; Kazemi et al., 2020; Wang et al., 2021b). In practice, it is often computationally infeasible to train and evaluate models on all possible edges one edge at a time. Thus, the chronologically ordered sequence of edges $E$ is usually divided into temporal batches $B_i^+$, where each batch has a fixed size of $b$ edges. Edges within the same batch are typically processed in parallel (Su et al., 2024; Rossi et al., 2020), thereby discarding temporal information within each batch. In addition to the existing (positive) edges $(u, v) \in B_i^+$, non-existing (negative) edges $(u^-, v^-) \in B_i^-$ are sampled and used for training and evaluation. This is done since real-world graphs are typically sparse and using all possible edges between all node pairs would lead to a large class imbalance and longer runtime.

While some TGNNs can utilize the edges' individual timestamps, e.g. via temporal encodings, sampling approaches for selecting recent neighbors (Rossi et al., 2020) or negative edges (Poursafaei et al., 2022) do not consider the temporal ordering of edges within batches. E.g. for negative sampling with collision checks specifically Poursafaei et al. (2022), this is because edges occurring as a positive sample in a batch cannot also be included as a negative sample in the same batch even with a different timestamp. Thus, while the prediction can utilize a positive sample's timestamp, this sample is ignored for the remaining batch duration during evaluation, essentially adopting the notion that edges within batches occur concurrently. With these assumptions, the task is formally defined as follows:

**Definition** (Dynamic link prediction).  Let $G = (V, E)$ be a temporal graph with node features $H_V$ and edge features $H_E$. Let $b$ be the batch size and $B_i^+ := \{(u, v) \mid \exists (u, v, t) \in E \text{ with } t \in \{t_{b \cdot i}, \ldots, t_{b \cdot (i+1)-1}\}\}$ the set of $b$ edges in the $i$-th batch. We further use $B_i^-$ to denote a set of negative edges drawn using negative sampling as described in Appendix A. For a given batch $i$ we use $\hat{E}_i = \{(u, v, t) \mid \exists (u, v, t) \in E : t < t_{b \cdot i}\}$ to denote the *past edges*. The goal of dynamic link prediction is to find a model $f_\theta\big(u, v \mid \hat{E}_i, H_V, H_{\hat{E}_i}\big)$ with parameters $\theta$ that, for all edges $(u, v) \in B_i^+ \cup B_i^-$ in each batch $i$, predicts whether $(u, v) \in B_i^+$ or $(u, v) \in B_i^-$.

**State-of-the-art TGNNs.**  Current state-of-the-art dynamic link prediction methods, such as JODIE (Kumar et al., 2019), DyRep (Trivedi et al., 2019), TGN (Rossi et al., 2020) keep an up-to-date memory of temporal information in the graph by utilizing recurrent neural networks. Temporal Graph Attention (TGAT) extends graph attention to the temporal domain and replaces positional encodings in GAT with a vector representation of time (Xu et al., 2020). TCL (Wang et al., 2021a) uses a transformer-based architecture to capture the nodes' time-evolving properties. CAWN learns temporal motifs based on causal anonymous walks (CAW) (Wang et al., 2021b). GraphMixer takes an attention-free and transformer-free approach, using an MLP-based link encoder, a mean-pooling-based node encoder, and an MLP-based link classifier for predictions (Cong et al., 2023). DyGFormer combines nodes' historical co-occurrences as interaction targets of the same source node with a temporal patching approach to capture long-term histories (Yu et al., 2023).

Several further approaches for discrete-time dynamic link prediction exist, including DyGEM (Taheri et al., 2019), DySAT (Sankar et al., 2020), and EvolveGCN (Pareja et al., 2020). For a recent survey of deep-learning-based dynamic link prediction, we refer to Feng et al. (2024).

**Temporal graph training.** Recent works (Su et al., 2024; Zhou et al., 2022; 2023) identified issues in the training setup for memory-based TGNNs with large batch sizes: Processing edges that belong to the same batch in parallel ignores their temporal dependencies, resulting in varying performance depending on the chosen batch size. This issue has been termed *temporal discontinuity*. Su et al. (2024) propose PRES which accounts for intra-batch temporal dependencies through a prediction-correction scheme. Zhou et al. (2023) propose a distributed framework using smaller batch sizes on multiple trainers. However, these works focus on training, not considering temporal discontinuity in evaluation.

**Temporal graph evaluation.** Recent progress in terms of TGNN evaluation includes the temporal graph benchmark (TGB) (Huang et al., 2023) similar to the static open graph benchmark (OGB) (Hu et al., 2020). Poursafaei et al. (2022) identify problems with random negative sampling for dynamic link prediction and propose new negative sampling techniques dependent on time to improve the evaluation of TGNNs. Gastinger et al. (2023) identify issues in the evaluation of temporal knowledge graph forecasting. Although none of the models used for this task overlap with regular TGNNs for dynamic link prediction, some of the problems can be related, e.g., differences in forecasting horizons leading to incomparable results.

## 3 FROM LINK PREDICTION TO LINK FORECASTING

Learning temporal patterns in a batch-oriented fashion leads to issues in continuous-time and discrete-time graphs. Below, we show that batching leads to inconsistent tasks because the time window for prediction varies for temporal batches across different link densities in time. Temporal batches further cause information loss or leakage by either inducing a non-existing temporal order between links or ignoring the existing order. We demonstrate these issues in eight continuous-time and six discrete-time temporal graphs, whose characteristics are summarized in Table 1 and Appendix B. To mitigate these issues, we then formulate the *link forecasting* task based on fixed-length time windows.

Table 1: Characteristics of continuous and discrete-time temporal graphs (Poursafaei et al., 2022; Yu et al., 2023). For each dataset, we list the type, the number of nodes $n$, the number of edges $m$, the resolution of timestamps, the total duration $T$ of the observation, the average number of edges $\overline{|E_t|}$ with the same timestamp $t$, and the temporal density $T/m$.

| Dataset | Type | $n$ | $m$ | Resolution | $T$ | $\overline{|E_t|}$ | $T/m$ |
|---------|------|-----|-----|------------|-----|--------------------|-------|
| Enron | Contin. | 184 | 125 235 | 1 second | 3.6 years | 5.5 ± 16.6 | 908.2 s |
| UCI | Contin. | 1899 | 59 835 | 1 second | 193.7 days | 1.0 ± 0.3 | 279.7 s |
| MOOC | Contin. | 7144 | 411 749 | 1 second | 29.8 days | 1.2 ± 0.5 | 6.2 s |
| Wiki. | Contin. | 9227 | 157 474 | 1 second | 31.0 days | 1.0 ± 0.2 | 17.0 s |
| LastFM | Contin. | 1980 | 1 293 103 | 1 second | 4.3 years | 1.0 ± 0.1 | 106.0 s |
| Myket | Contin. | 17 988 | 694 121 | 1 second | 197.0 days | 1.0 ± 0.0 | 24.5 s |
| Social | Contin. | 74 | 2 099 519 | 1 second | 242.3 days | 3.7 ± 2.5 | 10.0 s |
| Reddit | Contin. | 10 984 | 672 447 | 1 second | 31.0 days | 1.0 ± 0.1 | 4.0 s |
| UN V. | Discrete | 201 | 1 035 742 | 1 year | 71.0 years | 14 385.3 ± 7142.1 | 36.1 min |
| US L. | Discrete | 225 | 60 396 | 1 congress | 11.0 congr. | 5033.0 ± 92.4 | $1.8 \cdot 10^{-4}$ congr. |
| UN Tr. | Discrete | 255 | 507 497 | 1 year | 30.0 years | 15 859.3 ± 3830.8 | 32.1 min |
| Can. P. | Discrete | 734 | 74 478 | 1 year | 13.0 years | 5319.9 ± 1740.5 | 91.8 min |
| Flights | Discrete | 13 169 | 1 927 145 | 1 day | 121.0 days | 15 796.3 ± 4278.5 | 5.4 s |
| Cont. | Discrete | 692 | 2 426 279 | 5 minutes | 28.0 days | 300.9 ± 342.4 | 1.0 s |

### 3.1 PROBLEMS IN BATCH-BASED DYNAMIC LINK PREDICTION

One issue of batch-oriented temporal graph learning and dynamic link prediction is that activities in real-world temporal graphs are inhomogeneously distributed across time. In Figure 2 we show the temporal activity in terms of the number of time-stamped edges within a given time interval both for continuous-time and discrete-time temporal graphs. For continuous-time data, we used binning in six-hour intervals. The results show that most real-world temporal graphs have highly

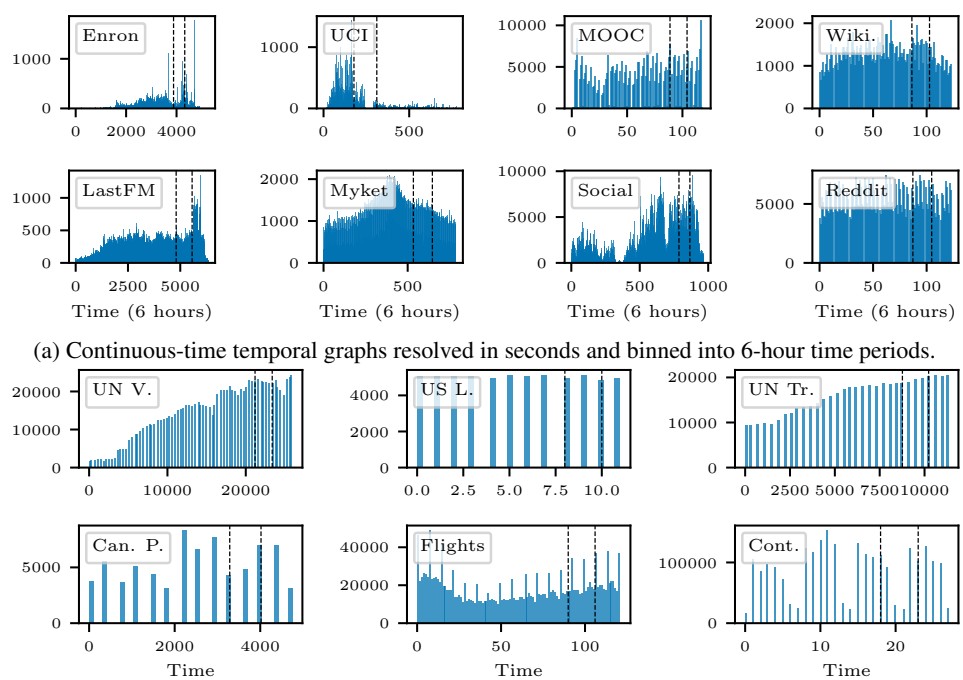

(a) Continuous-time temporal graphs resolved in seconds and binned into 6-hour time periods.

(b) Discrete-time temporal graphs with different time resolution (see Table 1).

Figure 2: Real-world datasets exhibit diverse edge occurrence patterns that are visualised using the edge density across time, i.e., histograms counting the number of edges per timestamp. Dashed lines divide the datasets into 70% train, 15% validation, and 15% test sets as used in Section 4.

inhomogeneous activities across time. For batch-oriented evaluation, this introduces the issue that each fixed-size batch $B_i^+$ determines a time window with duration $t_{b\cdot(i+1)-1} - t_{b\cdot i}$, i.e., shorter or longer during periods with higher or lower activity, respectively.

In Figure 3 we evaluate the dependency between batch size and time window for empirical temporal graphs. We observe that, both in continuous- and discrete-time temporal graphs, a single batch size can create time windows with varying durations even within the same dataset. For continuous-time temporal graphs, we typically have much bigger batches than edges per timestamp such that the time window defined by the batches become long (cf. Table 1). The number of edges per snapshot in discrete-time temporal graphs is generally larger than the batch size $b$ in any period regardless of the density (Table 1). This means that edges in a batch often belong to the same snapshot leading to small window durations.

As an example, consider the Myket dataset (Loghmani & Fazli, 2023) which contains users $v$ and Android applications $u$, connected at time $t$ when user $v$ installs application $u$. The timestamps are provided in seconds and edges occur roughly every 30 seconds on average (cf. Table 1), making the expected time range for a batch with size $b = 2$ approximately 30 seconds. With $b = 2$, the task is to predict which users install what applications during this time window. Choosing $b = 120$ or $b = 2880$ turns the task into a prediction problem for approximately the next hour or day, respectively. As we can see, batching not only leads to incomparable prediction tasks between models and datasets due to the varying window duration but also acts as a kind of coarse-graining discarding temporal information inside each batch.

In Figure 4 we use normalized mutual information (NMI) (Cover & Thomas, 2006) to measure the information loss caused by splitting the temporal edges into batches. NMI quantifies how much information observing one random variable conveys about another random variable (see Appendix G for more information). It takes values between 0, meaning "no information", and 1, meaning "full information". By treating the index $i$ of each batch $B_i$ assigned to each edge $(u, v) \in B_i$ as one random variable and the associated edge's timestamp $t$ as the other, we can measure the temporal information that is retained after dividing edges into batches. In this case, an NMI value of 1 means that we can reconstruct the timestamps of edges correctly from their batch number, and a value of 0

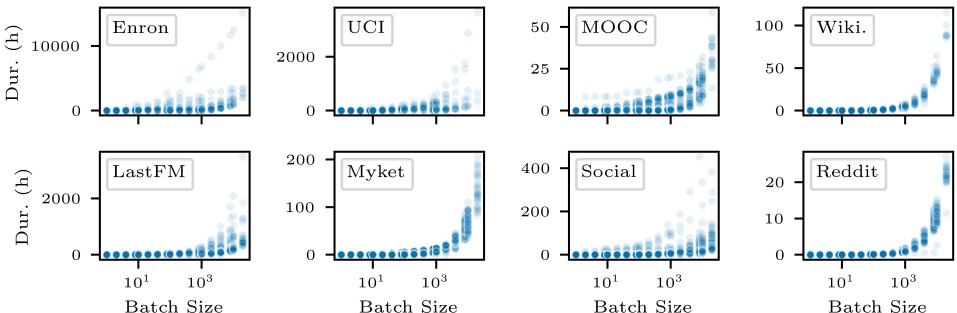

(a) Continuous-time temporal graphs: Batch size $b$ determines the average time window length. However, a single batch size creates time windows with various lengths within and across datasets.

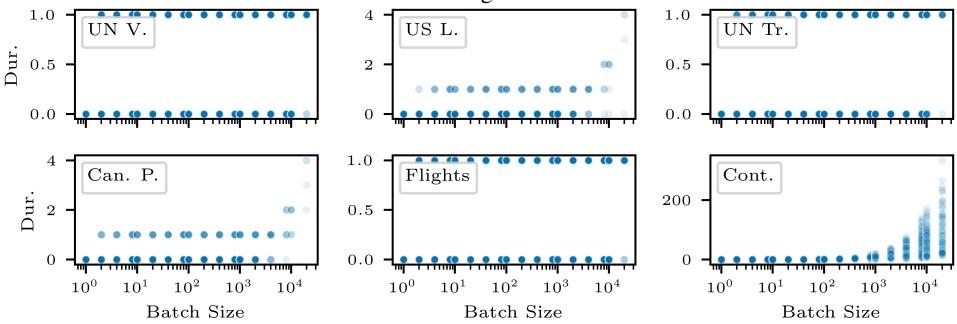

(b) Discrete-time temporal graphs: Fixed-size batches fall mostly within snapshots when the batches are much smaller than the snapshots. Depending on the dataset, larger batches can also span across many snapshots.

Figure 3: Using a low opacity value for individual points, the distribution of time window durations is shown for different batch sizes. I.e. points appear less see-through with an increasing number of points with the same duration and batch size stacked on top of each other.

means that batch numbers do not carry any information about timestamps. Consequently, small NMI values indicate a large loss of temporal information due to batching.

In Figure 4a we see that in continuous-time temporal graphs where timestamps have a high resolution, larger batches result in more information loss because assigning edges that occur at different times to the same batch discards their temporal ordering; the larger the batch size, the more information is lost. A batch size of $b = 1$ preserves most temporal information – i.e. maximum NMI – because we obtain a bijective mapping between almost all timestamps and batch numbers, except when multiple edges happen simultaneously.

Figure 4b shows the batch-size dependent NMI for discrete-time temporal graphs. The "optimal" batch size that retains most temporal information depends on the average number of links per snapshot and, thus, on the characteristics of the data. Too small batch sizes impose an ordering on the edges within the snapshots that is not present in the data while too large batches stretch across snapshots and discard the temporal ordering of edges from different batches. Additionally, information about the patterns inside each snapshot is leaked when edges with the same timestamp are evaluated sequentially in different batches, providing an unfair advantage for memory-based models that can utilize this information during inference.

**Link prediction vs. link forecasting:**   These results show that the aggregation of time-stamped edges within batches of varying duration loses information about the temporal ordering of interactions, but also introduces a non-existent order in snapshots larger than the batch size. This non-existent order can further lead to information leakage about patterns within snapshots. The results further highlight that changing the batch size influences both the prediction time window as well as the temporal information available to TGNNs. Effectively, the batch size is a hidden hyperparameter that directly impacts the characteristics (and difficulty) of the link prediction task. In real-world applications, however, the prediction time window is inherently connected to the problem at hand, necessitating a task formulation that is chosen carefully for each dataset instead of for each model. To address these issues, we propose a new *link forecasting* task that utilizes a fixed prediction time

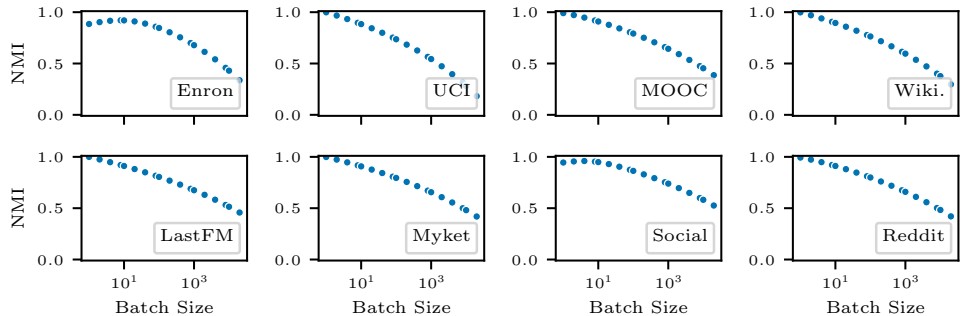

(a) For continuous-time, the temporal ordering of edges within batches is discarded. With increasing batch size, more edges with different timestamps are assigned to the same time window, thus, losing more information.

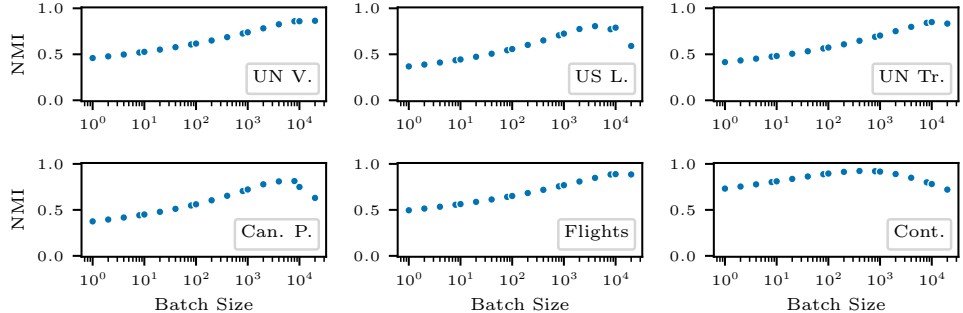

(b) Small batches for discrete-time temporal graphs implicitly define an edge ordering within snapshots that is not present in the data, consequently losing the information that edges of the same snapshot occur at the same time. The NMI has its maximum near the average snapshot size (refer to Table 1) after which the values decrease again similar to continuous-time datasets.

Figure 4: Temporal information loss in terms of Normalized Mutual Information (NMI, y-axis) for different batch sizes (x-axis), where smaller NMI scores indicate more information loss.

window with a variable number of edges. Compared to dynamic link prediction with a fixed batch size, this task is both easier and harder: It is easier because it limits the window durations in which the temporal ordering of edges is lost and, additionally, does not introduce artificial temporal orderings that are not present in the data. It is harder because it prevents information leakage and ensures that the task is not tuned to fit the model.

## 3.2 LINK FORECASTING: TASK DEFINITION

The study of temporal information is at the center of time series forecasting (Benidis et al., 2023) and, therefore, we relate our task definition to a fixed temporal quantity to solve the identified problems. We can interpret the temporal edges $E$ as $n^2$ Boolean time series, each of which takes the value 1 at the times an edge occurs. Standard multivariate models output a value for each timestamp over a forecasting horizon $h$. In large-scale temporal graphs, it is computationally infeasible to forecast the existence of all $n^2$ possible links, thus, only a sample of negative edges is considered instead. In continuous-time dynamic graphs, observations are available at high resolution, e.g. seconds, however, for many practical applications, predicting at lower granularity suffices. For example, it is typically enough to predict whether a customer purchases a certain product within the next day or week. Therefore, we consider forecasting for all timestamps $[t + 1, t + h]$ during a time window at once instead of for each of them individually, and define the link forecasting task as follows:

**Definition** (Dynamic link forecasting). Let $G = (V, E)$ be a temporal graph with node features $H_V$ and edge features $H_E$. Let $h$ be the time horizon and $W_i^+ := \{(u, v) \mid \exists (u, v, t) \in E \text{ with } i \cdot h \le t < (i + 1) \cdot h\}$ the set of edges in the $i$-th time window. We use $W_i^-$ to denote a sample of $|W_i^+|$ negative edges that are sampled using one of the negative sampling approaches described in appendix A and do not occur as positive edges in time window $[i \cdot h, (i + 1) \cdot h)$, i.e. $W_i^- \cap W_i^+ = \emptyset$. We further use $\hat{E}_i = \{(u, v, t) \mid \exists (u, v, t) \in E : t < i \cdot h\}$ to denote the set of past edges for time window $i$. The goal of dynamic link forecasting is to find a model $f_\theta(u, v | \hat{E}_i, H_V, H_{\hat{E}_i})$ with parameters $\theta$

that, for all edges $(u, v) \in W_i^+ \cup W_i^-$ in each time window $i$, forecasts whether $(u, v) \in W_i^+$ or $(u, v) \in W_i^-$.

Crucially, this definition makes the evaluation *independent of the batch size $b$* and instead introduces a time horizon $h$ that defines the forecasting time window. Essentially, this forecasting time window aggregates edges into snapshots, discarding temporal information inside each time window since we are forecasting whether a link exists in *any* of the timestamps $t \in [t + 1, t + h]$. Similar to batch-based dynamic link prediction, adopting such a time-window-based perspective enables parallel processing of edges, but does not preclude information loss entirely. Instead, time windows require deliberately choosing the temporal resolution based on the characteristics of each dataset and application (see appendix C for examples), resulting in a trade-off between evaluation runtime and granularity of temporal information available to the TGNN models. To summarize, link forecasting requires choosing a time horizon that affects and controls the information loss, but ensures that this information loss is consistent for different models, facilitating fair performance comparisons.

**Computational cost**  Let $m$ be the total number of links, where typically $m \ll n^2$ since real-world graphs are sparse, then we can assign each link to its corresponding time window in $O(m)$ by checking each link's interaction time. After each link is assigned, the time complexity during model evaluation is the same as for the batch-based approach. Since the number of links per time window varies and windows can become large during periods when many temporal edges occur, we cannot preclude memory overflows entirely. To mitigate this issue, one can split large time windows into smaller batches for GPU-based computations. Since information leakage instead of information loss needs to be prevented between batches of the same time window, steps such as negative sampling or memory updates need to be done based on the whole time window.

**Implementation**  We provide implementations for our evaluation procedure in commonly used PyTorch libraries to simplify the adoption of our approach. Specifically, we implement a new `DataLoader` called `SnapshotLoader` that replaces the widely used `TemporalDataLoader` in PyTorch Geometric (Fey & Lenssen, 2019). We extend DyGLib (Yu et al., 2023) with a command line argument `horizon` that can be used in the evaluation pipeline. The latter was used for the experiments in this work and can be used to reproduce our results. The implementations are added as supplementary material to ensure anonymity and will be made publicly available after acceptance of the paper.

## 4  LINK PREDICTION VS. FORECASTING IN STATE-OF-THE-ART TGNNs

We now experimentally evaluate the performance of nine state-of-the-art models (Kumar et al., 2019; Trivedi et al., 2019; Xu et al., 2020; Rossi et al., 2020; Wang et al., 2021b; Poursafaei et al., 2022; Wang et al., 2021a; Cong et al., 2023; Yu et al., 2023), both for the (conventional) dynamic link prediction task as well as our proposed dynamic link forecasting task. We use implementation and model configurations provided by DyGLib (Yu et al., 2023) (cf. Appendix D) and repeat each experiment five times to obtain averages. We use historical negative sampling (Poursafaei et al., 2022) and train each model using batch-based training and validation with batch size $b = 200$ which was found "to be a good trade-off between speed and update granularity" (Rossi et al., 2020) and adopted in similar works (Yu et al., 2023; Poursafaei et al., 2022). Afterwards, we evaluate each model trained with batch-based training using our proposed time-window-based evaluation method as well as the common batch-based evaluation approach with $b = 200$. We choose the forecasting horizon $h$ such that we obtain average batch sizes of approximately 200 for all continuous-time datasets to make the results of our new evaluation method comparable to the results of the batch-oriented approach (see Table 2 for the exact values). For discrete-time temporal graphs, we set $h = 1$ to obtain one time window per snapshot, predicting for time intervals ranging from five minutes to a year, depending on the datasets (cf. Table 1). Note that we choose $h$ based on the batch size $b = 200$ to achieve as much comparability as possible between the link forecasting and prediction tasks, thereby emphasizing performance differences between both approaches by "only" grouping the edges differently. We quantify how much information is shared between both approaches in Table 2 by calculating the NMI score between time window ID and batch ID as random variables. Additionally, we evaluate the models' performance based on realistic time horizons and present the results in Appendix C.

Table 2: Average links per window $|W_i^+|$ and standard deviation for horizon $h$ used in the evaluation (left). We chose $h$ for each dataset to get $\overline{|W_i^+|} \approx 200$. The average time window duration is provided in hours and seconds per batch (center). NMI uses the time window and batch IDs of the test set (cf. Appendix G) quantifying how much the chosen chunks differ between the approaches (right).

| Dataset | $h$ | $\overline{|W_i^+|}$ | $b$ | Avg Duration (h) | Avg Duration (s) | NMI |
|---|---|---|---|---|---|---|
| Enron | 172 800s (48h) | 214.1 ± 274.1 | 200 | 50.1 ± 165.38 | 180395.2 ± 595358.23 | 0.80 |
| UCI | 57 600s (16h) | 208.5 ± 335.5 | 200 | 15.4 ± 31.67 | 55542.5 ± 114021.12 | 0.83 |
| MOOC | 1200s (1/3h) | 199.3 ± 167.2 | 200 | 0.3 ± 0.72 | 1242.8 ± 2597.53 | 0.88 |
| Wikipedia | 3600s (1h) | 211.7 ± 56.3 | 200 | 0.9 ± 0.26 | 3382.2 ± 921.97 | 0.89 |
| LastFM | 21 600s (6h) | 204.4 ± 120.2 | 200 | 5.9 ± 5.37 | 21099.5 ± 19331.76 | 0.91 |
| Myket | 5400s (3/2h) | 220.1 ± 133.6 | 200 | 1.4 ± 1.18 | 4879.2 ± 4236.54 | 0.91 |
| Social Evo. | 1800s (1/2h) | 186.1 ± 165.3 | 200 | 0.6 ± 1.26 | 1984.1 ± 4533.72 | 0.91 |
| Reddit | 900s (1/4h) | 226.0 ± 54.5 | 200 | 0.2 ± 0.06 | 792.5 ± 199.35 | 0.92 |

The results are presented in Table 3 for continuous-time and in Table 4 for discrete-time temporal graphs. The tables show AUC-ROC scores for time-window-based link forecasting and the relative change compared to the batch-based evaluation of dynamic link prediction (average precision scores are provided in Appendix E). For continuous-time temporal graphs, the change in performance between our window-based and the batch-based approach largely depends on the dataset: Datasets with a similar window duration for all fixed-sized batches (quantified by NMI scores close to one in Table 2), such as Wikipedia, Reddit, or Myket, only exhibit small differences between the performances. This is expected since we chose the horizon $h$ to produce batches of the same average size as the fixed-sized batches. Nevertheless, we observe lower NMI values in Table 2 for datasets with inhomogeneously distributed temporal activity such as Enron or UCI – i.e. the time windows do not fit the fixed-sized batches well. These datasets with lower NMIs show substantial performance changes across models. This highlights that the performance scores of batch-based evaluation are skewed and may not reflect the models' performance in a real-world setting on inhomogeneous temporal datasets.

Table 3: Test AUC-ROC scores for link forecasting and the relative change compared to link prediction for continuous-time graphs on the *same trained models* (standard deviations in Appendix E). We compute the AUC-ROC score per time window and average by weighing each time window equally, regardless of the number of edges (Appendix F discusses additional weighting schemes). The last row/column provides mean $\mu$ and standard deviation $\sigma$ of the absolute values of the relative change per column/row.

| Dataset | JODIE | DyRep | TGN | TGAT | CAWN | EdgeBank | TCL | GraphMixer | DyGFormer | $\mu \pm \sigma$ |
|---|---|---|---|---|---|---|---|---|---|---|
| Enron | 84.0(↑8.6%) | 80.3(↑9.2%) | 67.9(↓0.2%) | 69.0(↑17.6%) | 75.7(↑13.9%) | 82.7(↑3.6%) | 75.1(↑11.1%) | 88.6(↑9.0%) | 84.5(↑10.6%) | 9.3%±5.1% |
| UCI | 86.8(↑4.2%) | 60.2(↑17.1%) | 62.1(↓1.5%) | 55.2(↑7.4%) | 56.5(↓3.0%) | 72.5(↑4.9%) | 56.3(↓6.2%) | 80.2(↓0.5%) | 75.7(↓0.6%) | 5.0%±5.1% |
| MOOC | 83.1(↓2.1%) | 79.0(↓2.1%) | 87.4(↓1.2%) | 79.9(↓2.9%) | 68.8(↓2.2%) | 59.8(↓3.4%) | 68.4(↓5.8%) | 70.3(↓5.5%) | 80.0(↓1.5%) | 3.0%±1.7% |
| Wiki. | 81.5(↓0.4%) | 78.3(↓0.1%) | 83.7(↓0.6%) | 82.9(↓0.7%) | 71.3(↓0.4%) | 77.2(↑0.1%) | 84.6(↓0.6%) | 87.3(↓0.6%) | 79.8(↓0.3%) | 0.4%±0.2% |
| LastFM | 76.3(↓2.2%) | 69.0(↓3.7%) | 79.2(↓1.9%) | 65.2(↓4.7%) | 66.3(↓2.6%) | 78.0(↓0.2%) | 62.5(↓2.7%) | 59.9(↓9.2%) | 78.2(↓1.0%) | 3.1%±2.6% |
| Myket | 64.4(↑0.6%) | 64.1(↓0.1%) | 61.2(↑0.1%) | 57.8(↑0.4%) | 33.5(↑3.1%) | 52.6(↑1.3%) | 58.2(↓0.3%) | 59.8(↑0.5%) | 33.8(↑3.0%) | 1.0%±1.2% |
| Social | 92.1(↑0.8%) | 92.2(↓0.5%) | 92.2(↑0.5%) | 92.5(↓0.1%) | 86.5(↓1.4%) | 84.9(↓1.1%) | 94.7(↓0.6%) | 94.6(↑0.6%) | 97.3(↑0.0%) | 0.6%±0.4% |
| Reddit | 80.6(↓0.0%) | 79.5(↑0.0%) | 80.4(↓0.0%) | 78.6(↓0.1%) | 80.2(↓0.0%) | 78.6(↓0.1%) | 76.2(↓0.1%) | 77.1(↓0.1%) | 80.2(0.0%) | 0.0%±0.1% |
| $\mu \pm \sigma$ | 2.4%±2.9% | 4.1%±6.1% | 0.8%±0.7% | 4.2%±6.0% | 3.3%±4.4% | 1.8%±1.9% | 3.4%±4.0% | 3.2%±4.0% | 2.1%±3.6% | |

Table 4: Test AUC-ROC scores as in Table 3 but for discrete-time graphs.

| Dataset | JODIE | DyRep | TGN | TGAT | CAWN | EdgeBank | TCL | GraphMixer | DyGFormer | $\mu \pm \sigma$ |
|---|---|---|---|---|---|---|---|---|---|---|
| UN V. | 54.0(↓26.7%) | 52.2(↓28.2%) | 51.3(↓27.1%) | 54.4(↑3.0%) | 53.7(↑7.1%) | 89.6(↑0.0%) | 53.4(↑0.6%) | 56.9(↑1.1%) | 65.2(↑3.5%) | 10.8%±12.6% |
| US L. | 52.5(↓6.8%) | 61.8(↓22.6%) | 57.7(↓31.2%) | 78.6(↑0.2%) | 82.0(↑0.2%) | 68.4(↑1.3%) | 75.4(↓0.3%) | 90.4(↑0.2%) | 89.4(↑0.0%) | 7.0%±11.7% |
| UN Tr. | 57.7(↓12.8%) | 50.3(↓20.4%) | 54.3(↓14.0%) | 64.1(↑3.9%) | 67.6(↑4.5%) | 85.6(↓1.0%) | 63.7(↑4.5%) | 68.6(↑3.4%) | 70.7(↑3.4%) | 7.5%±6.6% |
| Can. P. | 63.6(↓0.5%) | 67.5(↑1.2%) | 73.2(↓0.2%) | 72.7(↑1.5%) | 70.0(↑2.9%) | 63.2(↑0.4%) | 69.5(↑2.0%) | 80.7(↓0.6%) | 85.5(↓12.5%) | 2.4%±3.9% |
| Flights | 67.4(↓3.1%) | 66.0(↓4.3%) | 68.1(↓1.0%) | 72.6(↑0.0%) | 65.2(↑0.3%) | 74.6(↑0.0%) | 70.6(↑0.0%) | 70.7(↓0.0%) | 68.6(↓0.5%) | 1.0%±1.6% |
| Cont. | 95.6(↑0.1%) | 94.9(↓0.5%) | 96.6(↑0.5%) | 95.9(↑0.6%) | 86.7(↑4.1%) | 93.0(↑0.9%) | 95.7(↑1.7%) | 95.2(↑1.1%) | 97.7(↑0.6%) | 1.1%±1.2% |
| $\mu \pm \sigma$ | 8.3%±10.2% | 12.9%±12.2% | 12.3%±14.1% | 1.5%±1.6% | 3.2%±2.7% | 0.6%±0.5% | 1.5%±1.7% | 1.1%±1.2% | 3.4%±4.7% | |

We further observe that, for link forecasting, the performance of memory-based models (JODIE, DyRep, TGN) on discrete-time temporal graphs tends to decrease more than for other methods. This is expected since these models incorporate information about the present snapshot by updating their memory based on prior batches, which means using part of the snapshot's edges to predict its remaining edges. Our evaluation method prevents this information leakage, which explains the

substantial drop in performance.

For non-memory-based models, the performance tends to be better for link forecasting compared to link prediction on most discrete-time datasets. This is within our expectations because our time-window-based approach prevents, for $h = 1$, all batches that stretch across multiple snapshots. The batch-based approach can have overlapping batches which results in information loss because, when making predictions, only edges that occur *before* the current batch may be used. However, since batches stretching across snapshots contain edges from multiple snapshots, all edges belonging to those snapshots must not be used to make predictions – because they have not occurred *before* the batch. Therefore, in the case of overlapping batches, not all available information can be used to make predictions. For Contacts – the discrete-time dataset with the highest NMI score, we see the smallest changes in model performance. This demonstrates that the models' performance obtained through batch-oriented evaluation reflects the time-window-based performance more closely when a given batch size defines more homogeneous time windows. However, this is often not the case in real-world discrete-time temporal graphs with low granularity and large snapshots.

## 5 CONCLUSION

In this work, we considered issues associated with current evaluation practices for dynamic link prediction in temporal graphs. To address computational limitations, edges in the test set are split into fixed-size batches, making the task "too easy" but at the same time "too hard": "Too easy" because state-of-the-art approaches for dynamic link prediction have treated the batch size as a tunable parameter. Changing the batch size, however, changes the prediction task, resulting in incomparable results between different batch sizes. For discrete-time temporal graphs, multiple batches that include edges from the same timestamp further leak information that can be utilized by memory-based TGNNs. "Too hard" since in continuous-time temporal graphs, fixed-size batches create varying-length time windows that essentially lead to a different task for each duration, requiring the model to capture multiple task definitions at the same time. Furthermore, for edges within a batch, information regarding their temporal ordering is lost. In discrete-time temporal graphs where snapshots are typically larger than the batch size, batches additionally impose an ordering of edges not present in the data.

We solve these issues by formulating the *dynamic link forecasting* task. Dynamic link forecasting acknowledges the resolution at which temporal interaction data is recorded and explicitly considers a forecasting horizon corresponding to a prediction time window of a *fixed duration*. Depending on the dataset and problem setting, the horizon may span seconds, minutes, hours, or longer, but crucially, time windows always span the same length. We evaluated dynamic link forecasting performance of nine state-of-the-art temporal graph learning approaches on 14 real-world datasets, comparing it to the common dynamic link prediction evaluation. We find substantial differences, especially for memory-based TGNNs. We provide data loader implementations for PyTorch Geometric and DyGLib to facilitate practical applications of our evaluation approach.

**Limitations and Open Issues** Limitations of our work include that our reformulation of the dynamic link prediction task suggests time-window-based approaches for model training, which however goes beyond the scope of our paper. Furthermore, the metrics used in this work consider the problem of dynamic link prediction or forecasting as binary classification, using one negative sample for each positive edge. Other approaches (Huang et al., 2023; You et al., 2022) consider the problem as a ranking task (e.g. using the mean reciprocal rank (MRR)) and compare each positive edge against a large number of negative samples. These approaches typically use only one positive sample per batch which alleviates the information loss and provides a better estimate of the models' precision due to the large number of negative samples. In contrast, our window-based approach allows the parallel processing of many positive samples in each time window, leading to a considerably faster evaluation by making reasonable simplifications of the task. Specifically, it is enough to make predictions at a lower resolution than the data is available in for many tasks, e.g. it may not be required to predict whether a customer will purchase a certain product within the next second; making such a prediction for the next day or week may be sufficient. Combining the advantages of both, our time-window-based evaluation and the ranking-based tasks, is left for future work.

Lastly, there are no expected negative societal impacts that go beyond those of other foundational works in machine learning research.

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

# A    Negative sampling approaches

Dynamic link prediction is typically framed as a binary classification problem to predict class 1 for existing links during a certain time window and 0 otherwise. Due to the sparsity of most real-world graphs, it usually suffices to train and evaluate using all existing (positive) edges and a sample of non-existing (negative) edges out of all possible edges $V^2$. In static link prediction, negative edges are typically sampled randomly from $V^2$ without replacement but Poursafaei et al. (2022) showed that this technique is ill-suited for dynamic link prediction. One reason is rooted in the characteristics of temporal graphs where already-seen interactions tend to repeat several times during the observation period. To address this issue, Poursafaei et al. (2022) introduced negative sampling which we cover in the following.

Given a training set $E_{\text{train}}$ and test set $E_{\text{test}}$, each containing a sequence of edges $E_{t_{\min}:t_{\max}}$ in the temporal graph $G$, we can define the following commonly used sampling strategies for drawing negative samples $B_i^-$ for batch $B_i^+$ with $|B_i^+| = |B_i^-|$ Poursafaei et al. (2022); Yu et al. (2023).

- Random: Sample $B_i^-$ from $V^2$ without replacement. The subgraph corresponding to $B_i^+$ is assumed to be sparse, making it unlikely to sample a positive edge $e \in B_i$ as negative.

- Historic: Sample $B_i^-$ without replacement from all training edges $E_{\text{hist}} = E_{\text{train}} \setminus \{(v_j, u_j)|t_{b \cdot i} \leq t_j \leq t_{b \cdot (i+1)}\}$ except the ones appearing at the same time as the edges in $B_i^+$. If $|E_{\text{hist}}| < |B_i^+|$, draw the remaining edges randomly as described above.

- Inductive: Sample $B_i^-$ without replacement from all unseen test edges $E_{\text{ind}} = E_{\text{test}} \setminus (E_{\text{train}} \cup \{(v_j, u_j)|t_{b \cdot i} \leq t_j \leq t_{(i+1) \cdot b}\})$ except the ones appearing at the same time as the edges in $B_i^+$. If $|E_{\text{ind}}| < |B_i^+|$, draw the remaining edges randomly as described above.

Note that we leave out the validation set $E_{\text{val}}$ for simplicity. Negative edges for $E_{\text{val}}$ can be sampled as for $E_{\text{test}}$.

# B DATASETS

In this work, we use eight continuous-time and six discrete-time datasets, listed in Table 1. Here, we describe what systems were observed to create the datasets and plot the datasets' link count histograms in Figure 5 and window sizes for varying time horizons in Figure 6.

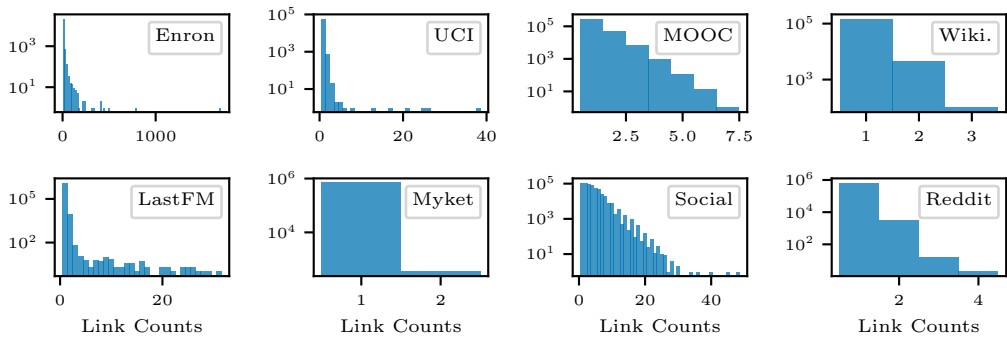

(a) In the continuous-time temporal graphs, it is most common that at most one edge occurs per timestamp.

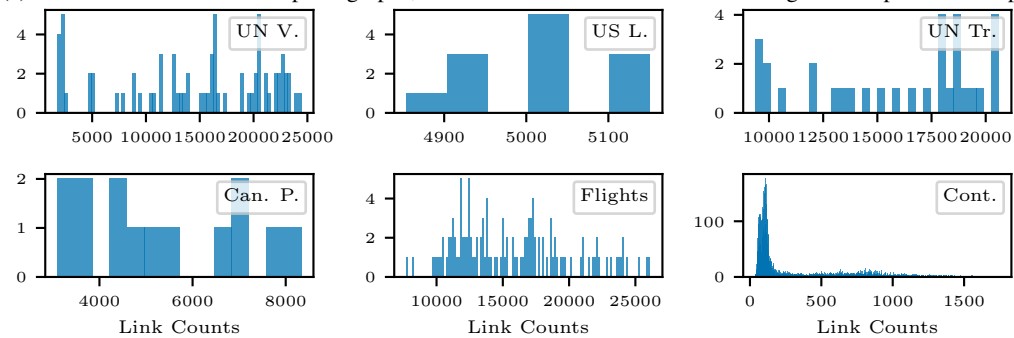

(b) The snapshots in discrete-time temporal graphs contain large numbers of edges, typically much larger than commonly utilized batch sizes. The Contacts dataset has fewer links per snapshot due to its much higher resolution than the remaining discrete-time datasets.

Figure 5: The link count histograms show how many edges occur per timestamp in continuous-time temporal graphs and per snapshot in discrete-time temporal graphs, respectively.

- **Enron** Shetty & Adibi (2004) is a bipartite continuous-time graph where nodes are users and the temporal edges represent emails sent between users. Emails with multiple recipients are recorded as separate and simultaneously occurring edges, one per recipient. The temporal edges are resolved at the second level and the dataset spans approximately 3.6 years.
- **UCI** Panzarasa et al. (2009) is a unipartite continuous-time social network dataset from an online platform at the University of California at Irvine. The nodes represent students and the timestamped edges represent communication between the students. The dataset spans approximately six and a half months.
- **MOOC** (massive open online course) Kumar et al. (2019) is a bipartite continuous-time graph where nodes represent users and units in an online course, such as problems or videos. Temporal edges are resolved at the second level and encode when a user interacts with a unit of the online course. The dataset spans approximately one month.
- **Wikipedia** (Wiki.) Kumar et al. (2019) is a bipartite continuous-time graph where nodes represent editors and Wikipedia articles. The timestamped edges are resolved at the second level and represent when an editor has edited an article. The dataset spans approximately one month.
- **LastFM** is a bipartite continuous-time graph where nodes represent users and songs. Temporal edges are resolved at the second level and model the users' listening behavior and represent when a user has listened to a song. The dataset was originally published by Celma (2010) and later filtered by Kumar et al. (2019) for use in a temporal graph learning context.

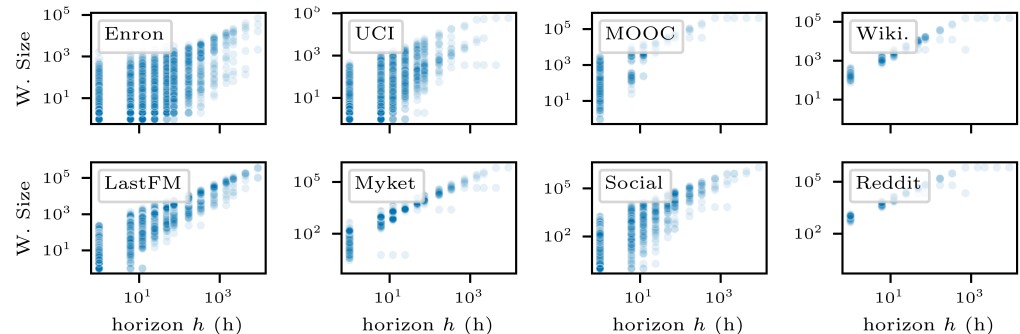

(a) The number of links per window in continuous-time temporal graphs for horizons ranging from one second to one year. The $x$-axis is labeled in hours.

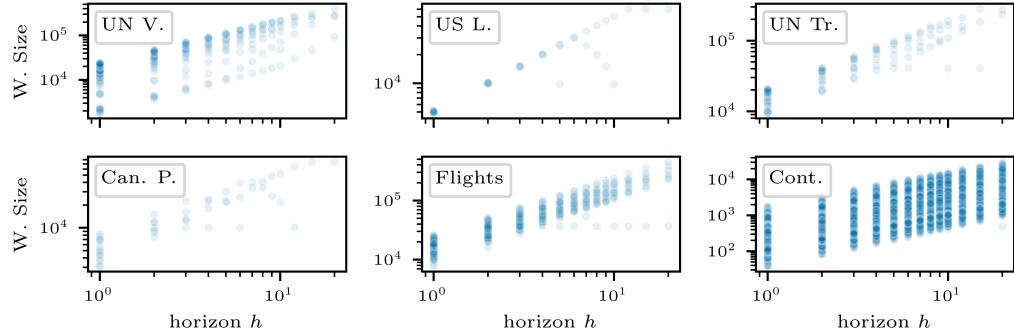

(b) Window sizes for discrete-time temporal graphs where $h$ represents the number of snapshots.

Figure 6: Window sizes, i.e. number of links per time window, for different horizons. For smaller values of $h$, we can see a wide range of window sizes for most datasets; for large values of $h$, i.e. horizons that include more than half of the observed time, the window sizes diverge because the window of the second time window that still includes observed links gets smaller.

- **Myket** Loghmani & Fazli (2023) is a bipartite continuous-time graph where nodes represent users and Android applications. The timestamped edges represent when a user installed an application. The dataset spans approximately six and a half months.

- **Social Evolution** (Social) Madan et al. (2012) is a unipartite continuous-time graph of the proximity between the students in a dormitory, collected between October 2008 and May 2009 using mobile phones. Temporal edges connect students when they are in proximity and are resolved in seconds.

- **Reddit** Kumar et al. (2019) is a bipartite continuous-time graph where nodes represent Reddit users and their posts. The timestamped edges are resolved in seconds and represent when a user has made a post on Reddit. The dataset spans approximately one month.

- **UN Vote** (UN V.)Voeten et al. (2009); Poursafaei et al. (2022) is a weighted unipartite discrete-time graph of votes in the United Nations General Assembly between 1946 and 2020. Nodes represent countries and edges connect countries if they both vote "yes". The dataset is resolved at the year level and edge weights represent how many times the two connected countries have both voted "yes" in the same vote.

- **US Legislators** (US L.) Huang et al. (2020); Fowler (2006); Poursafaei et al. (2022) is a weighted unipartite discrete-time graph of interactions between legislators in the US Senate. Nodes represent legislators and edges represent co-sponsorship, i.e., edges connect legislators who co-sponsor the same bill. The dataset is resolved at the congress level and edge weights encode the number of co-sponsorships during a congress.

- **UN Trade** (UN Tr.) MacDonald et al. (2015); Poursafaei et al. (2022) is a directed and weighted unipartite discrete-time graph of food and agricultural trade between countries where nodes represent countries. The dataset spans 30 years and is resolved at the year level. Weighted edges encode the sum of normalized agriculture imports or exports between two countries during a given year.

- **Canadian Parliament** (Can. P.) Huang et al. (2020); Poursafaei et al. (2022) is a weighted unipartite discrete-time political network where nodes represent Members of the Canadian Parliament (MPs) and an edge between two MPs means that they have both voted "yes" on a bill. The dataset is resolved at the year level and the edges' weights represent how often the two connected MPs voted "yes" on the same bill during a year.
- **Flights** Schäfer et al. (2014); Poursafaei et al. (2022) is a directed and weighted unipartite discrete-time graph where nodes represent airports and edges represent flights during the COVID-19 pandemic. The edges are resolved at the day level and their weights are given by the number of flights between two airports during the respective day.
- **Contacts** (Cont.) Sapiezynski et al. (2019); Poursafaei et al. (2022) is a unipartite discrete-time proximity network between university students. Nodes represent students who are connected by an edge if they were in close proximity during a time window. The dataset is resolved at the 5-minute level and spans one month.

## C  EVALUATION BASED ON REALISTIC TIME HORIZONS

In our main experiments, we determined the time horizon based on the average number of links per time window to enable a fair comparison to the batch-based approach. In section 3.1, we discussed that the time horizon for a realistic evaluation should instead be chosen carefully for each individual dataset. To provide an example of such a realistic evaluation, we will assign a reasonable horizon for each dataset used in this work. Note that this enables a fair model comparison since we use the same horizon for all models across each dataset. The temporal resolution for most discrete-time datasets is limited due to the data collection process. The US Legislators dataset for example contains one snapshot for each congress which provides a natural horizon of one snapshot. Thus, we will only consider continuous-time datasets and discrete-time datasets where the duration of a snapshot is not yet a natural time horizon for evaluation, i.e. Contacts.

We list the considered datasets and the chosen horizon with the reasoning behind it in the following:

- **Enron** is a network of users with edges representing emails sent between them. We choose 24 hours as a reasonable horizon since 90% of all email replies are typically sent within a day Kooti et al. (2015).

- **UCI** is a social network based on student communications. We use 30 min as the horizon since users receiving a text message typically feel pressured to reply between the next 20 minutes and the end of the day Aranda & Baig (2018).

- **MOOC** connects students to units of an online course based on their interactions. We set $h = 6$ min since Guo et al. (2014) recommend keeping the learning video of a unit shorter than this time frame.

- **Wikipedia** represents the editing behaviour of users in a graph. Since editing Wikipedia articles is unpaid, we do not expect frequent interactions from each user. We assume that users come from different time zones and consider that the total duration of the dataset is only one month. Therefore, we see the typical working time of 8 hours as an appropriate time horizon to take into account that interactions won't appear very frequently but there are still enough time windows for evaluation.

- **LastFM** connects users to the songs they listen to. We select 24 hours as the horizon to evaluate this dataset on a task where the goal is to predict the songs that users will listen to tomorrow based on their listening behaviour during the last days.

- **Myket** represents users and Android applications that are connected when an application is installed. Considering that, similar to the Wikipedia dataset, no frequent interactions are expected, we choose 24 hours as the time horizon since the total duration of the dataset is longer than the total duration of the Wikipedia dataset.

- **Social Evolution** is a proximity network gathered from students in a dormitory. Thus, we select 2 hours – a typical duration of a lecture including breaks – as the time horizon.

- **Reddit** contains the posting behaviour of Reddit users. Since dynamics in a social network are typically fast, we select a time horizon of 15 minutes.

- **Contacts:** Similar to Social Evolution, we select 2 hours as the time horizon since the network also captures the proximity of students.

We use the trained models from the experiments of the main part of our work and reevaluate the specified datasets with the selected time horizons. The results are presented in Table 5 and 6. For completeness, both tables also include the performance scores of the discrete-time datasets that have not been reevaluated because we determined the horizon used above as realistic.

The results using both the AUC-ROC as well as the average precision score mostly agree on a best-performing model, yet for different datasets, there is no clear winner among the models. For continuous-time datasets, JODIE, GraphMixer, and DyGFormer are among the best-performing models while EdgeBank, GraphMixer, and DyGFormer performed best for discrete-time data.

DyGFormer performs best for both Contacts and Social Evolution, suggesting that DyGFormer is best suited for proximity networks among all models. For UN Vote, UN Trade, and Flights, EdgeBank – a simple baseline model that predicts an edge if it has occurred before – is among the best. These datasets are highly repetitive because they are, e.g. based on a schedule or relations among countries

Table 5: Average AUC-ROC performance and standard deviation over five runs using the trained models from the above experiments and the window-based evaluation with the horizons specified above. The tables also include the datasets that have not been reevaluated using performance scores obtained with the time horizons of the main experiments. Due to time constraints, only one run using CAWN on the Contacts dataset finished in time. The score of the single run is reported below and will be replaced by the mean and standard deviation over five runs in the camera-ready version.

| Datasets | JODIE | DyRep | TGN | TGAT | CAWN | EdgeBank | TCL | GraphMixer | DyGFormer |
|---|---|---|---|---|---|---|---|---|---|
| Enron | 84.2 ± 4.8 | 80.4 ± 2.5 | 70.0 ± 4.5 | 68.3 ± 1.8 | 75.5 ± 0.4 | 83.1 ± 0.0 | 74.2 ± 4.8 | **87.9 ± 0.6** | 83.9 ± 0.6 |
| UCI | **89.4 ± 0.7** | 70.4 ± 2.8 | 64.0 ± 1.3 | 52.2 ± 1.1 | 56.0 ± 0.2 | 75.3 ± 0.0 | 55.3 ± 1.3 | 82.7 ± 0.9 | 75.4 ± 0.4 |
| MOOC | 84.6 ± 4.1 | 81.3 ± 3.3 | **87.6 ± 1.3** | 80.7 ± 0.9 | 69.3 ± 1.5 | 62.9 ± 0.0 | 68.1 ± 0.9 | 70.4 ± 1.6 | 79.9 ± 10.0 |
| Wiki. | 75.5 ± 0.8 | 73.2 ± 0.9 | 83.5 ± 0.6 | 83.5 ± 0.2 | 71.9 ± 0.8 | 76.1 ± 0.0 | 85.2 ± 0.6 | **87.9 ± 0.3** | 80.6 ± 1.6 |
| LastFM | 74.9 ± 2.1 | 67.5 ± 1.4 | 78.6 ± 2.9 | 66.0 ± 0.9 | 67.2 ± 0.3 | 77.5 ± 0.0 | 63.2 ± 6.7 | 60.6 ± 1.3 | **78.9 ± 0.6** |
| Myket | **64.5 ± 1.8** | 62.7 ± 3.1 | 60.4 ± 2.3 | 57.8 ± 0.4 | 32.2 ± 0.4 | 51.4 ± 0.0 | 58.3 ± 2.0 | 59.9 ± 0.3 | 32.8 ± 0.9 |
| Social | 87.0 ± 2.0 | 84.4 ± 4.7 | 92.3 ± 2.7 | 92.6 ± 0.5 | 86.7 ± 0.0 | 85.1 ± 0.0 | 94.7 ± 0.5 | 94.7 ± 0.2 | **97.4 ± 0.1** |
| Reddit | **80.6 ± 0.1** | 79.5 ± 0.8 | 80.4 ± 0.4 | 78.6 ± 0.7 | 80.2 ± 0.3 | 78.6 ± 0.0 | 76.2 ± 0.4 | 77.1 ± 0.4 | 80.2 ± 1.1 |
| UN V. | 54.0 ± 1.8 | 52.2 ± 2.0 | 51.3 ± 7.1 | 54.4 ± 3.6 | 53.7 ± 2.1 | **89.6 ± 0.0** | 53.4 ± 1.0 | 56.9 ± 1.6 | 65.2 ± 1.1 |
| US L. | 52.5 ± 1.8 | 61.8 ± 3.5 | 57.7 ± 1.8 | 78.6 ± 7.9 | 82.0 ± 4.0 | 68.4 ± 0.0 | 75.4 ± 5.3 | **90.4 ± 1.5** | 89.4 ± 0.9 |
| UN Tr. | 57.7 ± 3.3 | 50.3 ± 1.4 | 54.3 ± 1.5 | 64.1 ± 1.3 | 67.6 ± 1.2 | **85.6 ± 0.0** | 63.7 ± 1.6 | 68.6 ± 2.6 | 70.7 ± 2.6 |
| Can. P. | 63.6 ± 0.8 | 67.5 ± 8.5 | 73.2 ± 1.1 | 72.7 ± 2.2 | 70.0 ± 1.4 | 63.2 ± 0.0 | 69.5 ± 3.1 | 80.7 ± 0.9 | **85.5 ± 3.5** |
| Flights | 67.4 ± 2.0 | 66.0 ± 1.9 | 68.1 ± 1.7 | 72.6 ± 0.2 | 65.2 ± 1.8 | **74.6 ± 0.0** | 70.6 ± 0.1 | 70.7 ± 0.3 | 68.6 ± 1.3 |
| Cont. | 85.4 ± 0.5 | 74.5 ± 3.1 | 94.6 ± 0.6 | 96.0 ± 0.2 | 86.6 ± nan | 85.8 ± 0.0 | 95.7 ± 0.4 | 95.2 ± 0.2 | **97.8 ± 0.0** |

that rarely change. Thus, since they are all outperformed by simple baselines, none of the proposed TGNN models adequately address the task of these datasets, i.e. finding edges that do not follow the schedule or some other reoccurring pattern. For other types of datasets like communication (e.g. Enron or UCI) or user-interaction networks (e.g. Wikipedia or MOOC), no clear patterns are visible.

Table 6: Mean average precision scores and standard deviation following Table 5.

| Datasets | JODIE | DyRep | TGN | TGAT | CAWN | EdgeBank | TCL | GraphMixer | DyGFormer |
|---|---|---|---|---|---|---|---|---|---|
| Enron | 82.2 ± 4.8 | 80.0 ± 3.4 | 71.2 ± 4.2 | 72.6 ± 1.3 | 77.1 ± 0.3 | 81.6 ± 0.0 | 77.9 ± 2.6 | **89.2 ± 0.4** | 85.0 ± 0.6 |
| UCI | **93.0 ± 0.5** | 81.5 ± 1.9 | 78.0 ± 0.9 | 71.6 ± 0.7 | 73.3 ± 0.2 | 75.6 ± 0.0 | 73.2 ± 0.7 | 89.6 ± 0.5 | 85.2 ± 0.3 |
| MOOC | 84.1 ± 5.4 | 79.4 ± 3.8 | **87.4 ± 1.6** | 83.9 ± 0.8 | 73.8 ± 1.1 | 62.1 ± 0.0 | 75.8 ± 0.5 | 75.6 ± 0.9 | 82.6 ± 8.9 |
| Wiki. | 78.3 ± 1.2 | 76.4 ± 0.7 | 88.3 ± 0.4 | 88.0 ± 0.2 | 75.7 ± 1.1 | 72.4 ± 0.0 | 89.7 ± 0.4 | **91.3 ± 0.2** | 84.0 ± 1.2 |
| LastFM | 73.6 ± 2.0 | 64.8 ± 1.8 | 78.6 ± 3.7 | 73.0 ± 0.8 | 69.2 ± 0.5 | 72.9 ± 0.0 | 71.2 ± 6.7 | 71.0 ± 1.1 | **81.3 ± 0.9** |
| Myket | **62.9 ± 1.5** | 60.2 ± 1.9 | 61.1 ± 1.7 | 56.8 ± 0.4 | 44.9 ± 0.2 | 51.1 ± 0.0 | 57.8 ± 2.2 | 59.0 ± 0.2 | 44.4 ± 1.7 |
| Social | 82.5 ± 4.0 | 81.6 ± 5.7 | 94.0 ± 1.8 | 95.0 ± 0.3 | 85.8 ± 0.1 | 79.9 ± 0.0 | 96.2 ± 0.3 | 95.8 ± 0.2 | **97.8 ± 0.1** |
| Reddit | 80.1 ± 0.3 | 79.2 ± 0.9 | 80.6 ± 0.6 | 78.6 ± 1.0 | 81.3 ± 0.4 | 73.5 ± 0.0 | 76.5 ± 0.6 | 77.5 ± 0.5 | **82.8 ± 0.8** |
| UN V. | 52.6 ± 1.8 | 49.6 ± 1.9 | 49.7 ± 3.9 | 52.7 ± 2.6 | 52.4 ± 2.0 | **84.2 ± 0.0** | 52.4 ± 0.9 | 54.0 ± 1.4 | 62.4 ± 1.7 |
| US L. | 46.0 ± 0.9 | 62.5 ± 3.6 | 58.6 ± 2.4 | 71.0 ± 8.9 | 80.7 ± 3.7 | 63.2 ± 0.0 | 77.5 ± 4.3 | **86.5 ± 1.9** | 86.1 ± 1.0 |
| UN Tr. | 52.7 ± 3.0 | 49.4 ± 0.9 | 53.2 ± 1.5 | 59.1 ± 2.7 | 59.2 ± 1.7 | **79.0 ± 0.0** | 57.5 ± 1.9 | 65.8 ± 1.9 | 67.1 ± 2.7 |
| Can. P. | 52.1 ± 0.5 | 61.0 ± 7.6 | 69.9 ± 0.8 | 70.8 ± 1.6 | 68.3 ± 2.3 | 59.4 ± 0.0 | 68.2 ± 1.6 | 80.9 ± 0.5 | **83.2 ± 2.9** |
| Flights | 65.2 ± 2.7 | 63.9 ± 2.8 | 68.3 ± 2.2 | **73.5 ± 0.3** | 64.7 ± 0.9 | 70.4 ± 0.0 | 71.0 ± 0.4 | 71.9 ± 0.8 | 68.9 ± 2.0 |
| Cont. | 83.3 ± 0.7 | 69.7 ± 3.7 | 95.0 ± 0.8 | 96.9 ± 0.2 | 88.3 ± nan | 82.3 ± 0.0 | 96.7 ± 0.5 | 95.8 ± 0.1 | **98.5 ± 0.0** |

# D EXPERIMENTAL DETAILS

For reproducibility, we provide a Python package extending the dynamic graph learning library DyGLib[1] Yu et al. (2023) as a supplement, including a bash script to run the experiments. The code will be made publicly available on GitHub after acceptance of the paper.

We use the best hyperparameters reported by Yu et al. (2023) and, for completeness, list these hyperparameters for the 13 datasets used by Yu et al. (2023) below. However, the Myket dataset Loghmani & Fazli (2023) was not included in the study. Therefore, for Myket, we use each method's default parameters as suggested by the respective authors.

We use 9 state-of-the-art dynamic graph learning models and baselines (JODIE Kumar et al. (2019), DyRep Trivedi et al. (2019), TGAT Xu et al. (2020), TGN Rossi et al. (2020), CAWN Wang et al. (2021b), EdgeBank Poursafaei et al. (2022), TCL Wang et al. (2021a), GraphMixer Cong et al. (2023) and DyGFormer Yu et al. (2023)). The neural-network-based approaches (all except EdgeBank) are trained five times for 100 epochs using the Adam optimizer with a learning rate of 0.0001. An early-stopping strategy with a patience of 5 is employed to avoid overfitting. For training and validation, a batch size of 200 is used. The training, validation and test sets of each dataset contain 70%, 15% and 15% of the edges, respectively. The sets are split based on time, i.e., the training set contains the edges that occurred first while the test set comprises the most recent edges.

The experiments were conducted on a variety of machines with different CPUs and GPUs. A list of machine specifications is provided in Table 7.

Table 7: Hardware details of the machines used for the experiments.

|  (a) CPUs  |  (b) GPUs  |
| --- | --- |
| **CPU** | **GPU** |
| AMD Ryzen Threadripper PRO 5965WX 24 Cores | NVIDIA GeForce RTX 3090 Ti |
| AMD Ryzen 9 7900X 12 Cores | NVIDIA GeForce RTX 4080 |
| 11th Gen Intel(R) Core(TM) i9-11900K 8 Cores | NVIDIA GeForce RTX 3090 |
| AMD Ryzen 9 7950X 16 Cores | NVIDIA GeForce RTX 4090 |
| 13th Gen Intel(R) Core(TM) i9-13900H 14 Cores | NVIDIA GeForce RTX 4060 (Laptop) |
|  | NVIDIA A100 |
|  | NVIDIA GeForce RTX 2080 Ti |
|  | NVIDIA TITAN Xp |
|  | NVIDIA TITAN X |
|  | NVIDIA Quadro RTX 8000 |

For all model architectures, time-related representations use a size of 100 dimensions while all other non-time-related representations are set to 172. An exception is DyGFormer where the neighbor co-occurrence encoding and the aligned encoding each have 50 dimensions. We use eight attention heads for CAWN, and two attention heads for all other attention-based methods. The memory-based models either use a vanilla recurrent neural network (JODIE and DyRep), or a gated recurrent unit (GRU) to update their memory. Other model-specific parameters are provided in Table 8 .

---

[1] https://github.com/yule-BUAA/DyGLib (MIT License)

Table 8: Specific hyperparameters for different models and datasets.

(a) Hyperparameters for neighborhood sampling-based models. $n_{\text{Neighbors}}$ is the number of sampled neighbors using the specified neighbor sampling strategy. $n_{\text{Layers}}$ is the number of transformer layers (for TCL), the number of MLP-Mixer layers (for GraphMixer) or the number of GNN layers otherwise.

| Dataset | Model | Neigh. Sampling | $n_{\text{Neighbors}}$ | $n_{\text{Layers}}$ | Dropout |
|---|---|---|---|---|---|
| | DyRep | recent | 10 | 1 | 0.1 |
| | TGAT | recent | 20 | 2 | 0.1 |
| Wikipedia | TGN | recent | 10 | 1 | 0.1 |
| | TCL | recent | 20 | 2 | 0.1 |
| | GraphMixer | recent | 30 | 2 | 0.5 |
| | DyRep | recent | 10 | 1 | 0.1 |
| | TGAT | uniform | 20 | 2 | 0.1 |
| Reddit | TGN | recent | 10 | 1 | 0.1 |
| | TCL | uniform | 20 | 2 | 0.1 |
| | GraphMixer | recent | 10 | 2 | 0.5 |
| | DyRep | recent | 10 | 1 | 0.0 |
| | TGAT | recent | 20 | 2 | 0.1 |
| MOOC | TGN | recent | 10 | 1 | 0.2 |
| | TCL | recent | 20 | 2 | 0.1 |
| | GraphMixer | recent | 20 | 2 | 0.4 |
| | DyRep | recent | 10 | 1 | 0.0 |
| | TGAT | recent | 20 | 2 | 0.1 |
| LastFM | TGN | recent | 10 | 1 | 0.3 |
| | TCL | recent | 20 | 2 | 0.1 |
| | GraphMixer | recent | 10 | 2 | 0.0 |
| | DyRep | recent | 10 | 1 | 0.0 |
| | TGAT | recent | 20 | 2 | 0.2 |
| Enron | TGN | recent | 10 | 1 | 0.0 |
| | TCL | recent | 20 | 2 | 0.1 |
| | GraphMixer | recent | 20 | 2 | 0.5 |
| | DyRep | recent | 10 | 1 | 0.1 |
| | TGAT | recent | 20 | 2 | 0.1 |
| Social Evo. | TGN | recent | 10 | 1 | 0.0 |
| | TCL | recent | 20 | 2 | 0.0 |
| | GraphMixer | recent | 20 | 2 | 0.3 |
| | DyRep | recent | 10 | 1 | 0.0 |
| | TGAT | recent | 20 | 2 | 0.1 |
| UCI | TGN | recent | 10 | 1 | 0.1 |
| | TCL | recent | 20 | 2 | 0.0 |
| | GraphMixer | recent | 20 | 2 | 0.4 |
| | DyRep | recent | 10 | 1 | 0.1 |
| | TGAT | recent | 20 | 2 | 0.1 |
| Myket | TGN | recent | 10 | 1 | 0.1 |
| | TCL | recent | 20 | 2 | 0.1 |
| | GraphMixer | recent | 20 | 2 | 0.1 |
| | DyRep | recent | 10 | 1 | 0.1 |
| | TGAT | recent | 20 | 2 | 0.1 |
| Flights | TGN | recent | 10 | 1 | 0.1 |
| | TCL | recent | 20 | 2 | 0.1 |
| | GraphMixer | recent | 20 | 2 | 0.2 |
| | DyRep | uniform | 10 | 1 | 0.0 |
| | TGAT | uniform | 20 | 2 | 0.2 |
| Can. Parl. | TGN | uniform | 10 | 1 | 0.3 |
| | TCL | uniform | 20 | 2 | 0.2 |
| | GraphMixer | uniform | 20 | 2 | 0.2 |
| | DyRep | recent | 10 | 1 | 0.0 |
| | TGAT | recent | 20 | 2 | 0.1 |
| US Legis. | TGN | recent | 10 | 1 | 0.1 |
| | TCL | uniform | 20 | 2 | 0.3 |
| | GraphMixer | recent | 20 | 2 | 0.4 |
| | DyRep | recent | 10 | 1 | 0.1 |
| | TGAT | uniform | 20 | 2 | 0.1 |
| UN Trade | TGN | recent | 10 | 1 | 0.2 |
| | TCL | uniform | 20 | 2 | 0.0 |
| | GraphMixer | uniform | 20 | 2 | 0.1 |
| | DyRep | recent | 10 | 1 | 0.1 |
| | TGAT | recent | 20 | 2 | 0.2 |
| UN Vote | TGN | uniform | 10 | 1 | 0.1 |
| | TCL | uniform | 20 | 2 | 0.0 |
| | GraphMixer | uniform | 20 | 2 | 0.0 |
| | DyRep | recent | 10 | 1 | 0.0 |
| | TGAT | recent | 20 | 2 | 0.1 |
| Contacts | TGN | recent | 10 | 1 | 0.1 |
| | TCL | recent | 20 | 2 | 0.0 |
| | GraphMixer | recent | 20 | 2 | 0.1 |

(b) Hyperparameters DyGFormer.

| Dataset | Model | Sequence Length | Patch Size | Dropout |
|---|---|---|---|---|
| Wikipedia | DyGFormer | 32 | 1 | 0.1 |
| Reddit | DyGFormer | 64 | 2 | 0.2 |
| MOOC | DyGFormer | 256 | 8 | 0.1 |
| LastFM | DyGFormer | 512 | 16 | 0.1 |
| Enron | DyGFormer | 256 | 8 | 0.0 |
| Social Evo. | DyGFormer | 32 | 1 | 0.1 |
| UCI | DyGFormer | 32 | 1 | 0.1 |
| Myket | DyGFormer | 32 | 1 | 0.1 |
| Flights | DyGFormer | 256 | 8 | 0.1 |
| Can. Parl. | DyGFormer | 2048 | 64 | 0.1 |
| US Legis. | DyGFormer | 256 | 8 | 0.0 |
| UN Trade | DyGFormer | 256 | 8 | 0.0 |
| UN Vote | DyGFormer | 128 | 4 | 0.2 |
| Contacts | DyGFormer | 32 | 1 | 0.0 |

(c) Hyperparameters CAWN.

| Dataset | Model | Walk Length | Time Scale | Dropout |
|---|---|---|---|---|
| Wikipedia | CAWN | 1 | 0.000001 | 0.1 |
| Reddit | CAWN | 1 | 0.000001 | 0.1 |
| MOOC | CAWN | 1 | 0.000001 | 0.1 |
| LastFM | CAWN | 1 | 0.000001 | 0.1 |
| Enron | CAWN | 1 | 0.000001 | 0.1 |
| Social Evo. | CAWN | 1 | 0.000001 | 0.1 |
| UCI | CAWN | 1 | 0.000001 | 0.1 |
| Myket | CAWN | 1 | 0.000001 | 0.1 |
| Flights | CAWN | 1 | 0.000001 | 0.1 |
| Can. Parl. | CAWN | 1 | 0.000001 | 0.0 |
| US Legis. | CAWN | 1 | 0.000001 | 0.1 |
| UN Trade | CAWN | 1 | 0.000001 | 0.1 |
| UN Vote | CAWN | 1 | 0.000001 | 0.1 |
| Contacts | CAWN | 1 | 0.000001 | 0.1 |

(d) Hyperparameters EdgeBank

| Dataset | Model | Neg. Sampling | Memory Mode | Time Window |
|---|---|---|---|---|
| | | random | unlimited | - |
| Wikipedia | EdgeBank | historical | repeat threshold | - |
| | | inductive | repeat threshold | - |
| | | random | unlimited | - |
| Reddit | EdgeBank | historical | repeat threshold | - |
| | | inductive | repeat threshold | - |
| | | random | time window | fixed proportion |
| MOOC | EdgeBank | historical | time window | repeat interval |
| | | inductive | repeat threshold | - |
| | | random | time window | fixed proportion |
| LastFM | EdgeBank | historical | time window | repeat interval |
| | | inductive | repeat threshold | - |
| | | random | time window | fixed proportion |
| Enron | EdgeBank | historical | time window | repeat interval |
| | | inductive | repeat threshold | - |
| | | random | repeat threshold | - |
| Social Evo. | EdgeBank | historical | repeat threshold | - |
| | | inductive | repeat threshold | - |
| | | random | unlimited | - |
| UCI | EdgeBank | historical | time window | fixed proportion |
| | | inductive | time window | repeat interval |
| | | random | unlimited | - |
| Myket | EdgeBank | historical | repeat threshold | |
| | | inductive | repeat threshold | |
| | | random | unlimited | - |
| Flights | EdgeBank | historical | repeat threshold | - |
| | | inductive | repeat threshold | - |
| | | random | time window | fixed proportion |
| Can. Parl. | EdgeBank | historical | time window | fixed proportion |
| | | inductive | repeat threshold | - |
| | | random | time window | fixed proportion |
| US Legis. | EdgeBank | historical | time window | fixed proportion |
| | | inductive | time window | fixed proportion |
| | | random | time window | repeat interval |
| UN Trade | EdgeBank | historical | time window | repeat interval |
| | | inductive | repeat threshold | - |
| | | random | time window | repeat interval |
| UN Vote | EdgeBank | historical | time window | repeat interval |
| | | inductive | time window | repeat interval |
| | | random | time window | repeat interval |
| Contacts | EdgeBank | historical | time window | repeat interval |
| | | inductive | repeat threshold | - |

# E    DETAILED AUC-ROC AND AVERAGE PRECISION RESULTS

Here, we provide detailed tabulated results for all models' AUC-ROC and average precision performance across five runs, including standard deviations.

## AUC-ROC

Table 9: Average AUC-ROC performance over five runs for the test set of the continuous-time datasets from Poursafaei et al. (2022); Yu et al. (2023), including standard deviations.

| Eval | Dataset | JODIE | DyRep | TGN | TGAT | CAWN | EdgeBank | TCL | GraphMixer | DyGFormer |
|---|---|---|---|---|---|---|---|---|---|---|
| Forec. | Enron | 84.0 ± 5.1 | 80.3 ± 1.4 | 67.9 ± 7.1 | 69.0 ± 1.6 | 75.7 ± 0.5 | 82.7 ± 0.0 | 75.1 ± 5.2 | 88.6 ± 0.5 | 84.5 ± 0.6 |
| | UCI | 86.8 ± 1.0 | 60.2 ± 2.8 | 62.1 ± 1.3 | 55.2 ± 1.4 | 56.5 ± 0.5 | 72.5 ± 0.0 | 56.3 ± 1.0 | 80.2 ± 1.0 | 75.7 ± 0.5 |
| | MOOC | 83.1 ± 4.2 | 79.0 ± 4.5 | 87.4 ± 1.9 | 79.9 ± 0.8 | 68.8 ± 1.6 | 59.8 ± 0.0 | 68.4 ± 1.4 | 70.3 ± 1.2 | 80.0 ± 9.0 |
| | Wiki. | 81.5 ± 0.4 | 78.3 ± 0.4 | 83.7 ± 0.6 | 82.9 ± 0.3 | 71.3 ± 0.8 | 77.2 ± 0.0 | 84.6 ± 0.5 | 87.3 ± 0.3 | 79.8 ± 1.6 |
| | LastFM | 76.3 ± 0.8 | 69.0 ± 1.4 | 79.2 ± 2.7 | 65.2 ± 0.9 | 66.3 ± 0.3 | 78.0 ± 0.0 | 62.5 ± 6.4 | 59.9 ± 1.4 | 78.2 ± 0.6 |
| | Myket | 64.4 ± 2.2 | 64.1 ± 2.9 | 61.2 ± 2.6 | 57.8 ± 0.5 | 33.5 ± 0.4 | 52.6 ± 0.0 | 58.2 ± 2.2 | 59.8 ± 0.4 | 33.8 ± 0.9 |
| | Social | 92.1 ± 1.9 | 92.2 ± 0.7 | 92.2 ± 2.6 | 92.5 ± 0.5 | 86.5 ± 0.0 | 84.9 ± 0.0 | 94.7 ± 0.5 | 94.6 ± 0.2 | 97.3 ± 0.1 |
| | Reddit | 80.6 ± 0.1 | 79.5 ± 0.8 | 80.4 ± 0.4 | 78.6 ± 0.7 | 80.2 ± 0.3 | 78.6 ± 0.0 | 76.2 ± 0.4 | 77.1 ± 0.4 | 80.2 ± 1.1 |
| Pred. | Enron | 77.4 ± 3.6 | 73.5 ± 2.4 | 68.0 ± 2.9 | 58.7 ± 1.2 | 66.4 ± 0.4 | 79.8 ± 0.0 | 67.6 ± 5.5 | 81.3 ± 0.8 | 76.4 ± 0.5 |
| | UCI | 83.3 ± 1.4 | 51.4 ± 7.8 | 63.0 ± 1.3 | 59.6 ± 1.5 | 58.2 ± 0.6 | 69.1 ± 0.0 | 60.0 ± 0.9 | 80.6 ± 0.8 | 76.2 ± 0.6 |
| | MOOC | 84.8 ± 3.1 | 80.7 ± 3.2 | 88.5 ± 1.6 | 82.3 ± 0.6 | 70.4 ± 1.3 | 61.9 ± 0.0 | 72.6 ± 0.6 | 74.4 ± 1.4 | 81.2 ± 8.9 |
| | Wiki. | 81.8 ± 0.4 | 78.4 ± 0.4 | 84.1 ± 0.6 | 83.5 ± 0.2 | 71.6 ± 0.8 | 77.1 ± 0.0 | 85.2 ± 0.5 | 87.8 ± 0.3 | 80.0 ± 1.6 |
| | LastFM | 78.0 ± 0.7 | 71.7 ± 1.1 | 80.7 ± 2.4 | 68.4 ± 0.7 | 68.1 ± 0.3 | 78.2 ± 0.0 | 64.3 ± 6.1 | 65.9 ± 1.7 | 78.9 ± 0.6 |
| | Myket | 64.0 ± 2.1 | 64.2 ± 2.7 | 61.1 ± 2.6 | 57.6 ± 0.4 | 32.5 ± 0.4 | 51.9 ± 0.0 | 58.4 ± 2.0 | 59.5 ± 0.4 | 32.8 ± 1.0 |
| | Social | 91.4 ± 2.1 | 92.7 ± 0.5 | 91.7 ± 3.3 | 92.6 ± 0.5 | 87.7 ± 0.1 | 85.8 ± 0.0 | 95.2 ± 0.2 | 94.1 ± 0.2 | 97.3 ± 0.1 |
| | Reddit | 80.6 ± 0.1 | 79.5 ± 0.8 | 80.4 ± 0.4 | 78.7 ± 0.6 | 80.2 ± 0.3 | 78.6 ± 0.0 | 76.2 ± 0.4 | 77.1 ± 0.4 | 80.2 ± 1.1 |

Table 10: Average AUC-ROC performance over five runs for the test set of the discrete-time datasets from Poursafaei et al. (2022); Yu et al. (2023), including standard deviation.

| Eval | Dataset | JODIE | DyRep | TGN | TGAT | CAWN | EdgeBank | TCL | GraphMixer | DyGFormer |
|---|---|---|---|---|---|---|---|---|---|---|
| Forec. | UN V. | 54.0 ± 1.8 | 52.2 ± 2.0 | 51.3 ± 7.1 | 54.4 ± 3.6 | 53.7 ± 2.1 | 89.6 ± 0.0 | 53.4 ± 1.0 | 56.9 ± 1.6 | 65.2 ± 1.1 |
| | US L. | 52.5 ± 1.8 | 61.8 ± 3.5 | 57.7 ± 1.8 | 78.6 ± 7.9 | 82.0 ± 4.0 | 68.4 ± 0.0 | 75.4 ± 5.3 | 90.4 ± 1.5 | 89.4 ± 0.9 |
| | UN Tr. | 57.7 ± 3.3 | 50.3 ± 1.4 | 54.3 ± 1.5 | 64.1 ± 1.3 | 67.6 ± 1.2 | 85.6 ± 0.0 | 63.7 ± 1.6 | 68.6 ± 2.6 | 70.7 ± 2.6 |
| | Can. P. | 63.6 ± 0.8 | 67.5 ± 8.5 | 73.2 ± 1.1 | 72.7 ± 2.2 | 70.0 ± 1.4 | 63.2 ± 0.0 | 69.5 ± 3.1 | 80.7 ± 0.9 | 85.5 ± 3.5 |
| | Flights | 67.4 ± 2.0 | 66.0 ± 1.9 | 68.1 ± 1.7 | 72.6 ± 0.2 | 65.2 ± 1.8 | 74.6 ± 0.0 | 70.6 ± 0.1 | 70.7 ± 0.3 | 68.6 ± 1.3 |
| | Cont. | 95.6 ± 0.8 | 94.9 ± 0.3 | 96.6 ± 0.3 | 95.9 ± 0.2 | 86.7 ± 0.1 | 93.0 ± 0.0 | 95.7 ± 0.5 | 95.2 ± 0.2 | 97.7 ± 0.0 |
| Pred. | UN V. | 73.7 ± 2.4 | 72.6 ± 1.5 | 70.3 ± 4.3 | 52.8 ± 3.6 | 50.1 ± 1.6 | 89.5 ± 0.0 | 53.0 ± 1.6 | 56.2 ± 2.0 | 63.0 ± 1.1 |
| | US L. | 56.3 ± 1.9 | 79.9 ± 1.1 | 84.0 ± 2.2 | 78.5 ± 7.8 | 81.8 ± 4.0 | 67.5 ± 0.0 | 75.6 ± 5.4 | 90.2 ± 1.6 | 89.4 ± 0.9 |
| | UN Tr. | 66.1 ± 3.0 | 63.2 ± 2.1 | 63.1 ± 1.2 | 61.7 ± 1.3 | 64.7 ± 1.3 | 86.4 ± 0.0 | 60.9 ± 1.3 | 66.3 ± 2.5 | 68.3 ± 2.3 |
| | Can. P. | 63.9 ± 0.7 | 66.6 ± 2.5 | 73.4 ± 3.5 | 71.6 ± 2.6 | 68.0 ± 1.0 | 62.9 ± 0.0 | 68.2 ± 3.6 | 81.2 ± 1.0 | 97.7 ± 0.7 |
| | Flights | 69.5 ± 2.2 | 69.0 ± 1.0 | 68.8 ± 1.6 | 72.6 ± 0.2 | 65.0 ± 1.4 | 74.6 ± 0.0 | 70.6 ± 0.1 | 70.7 ± 0.3 | 68.9 ± 1.1 |
| | Cont. | 95.5 ± 0.6 | 95.4 ± 0.2 | 96.1 ± 0.8 | 95.4 ± 0.3 | 83.3 ± 0.0 | 92.2 ± 0.0 | 94.1 ± 0.8 | 94.1 ± 0.2 | 97.1 ± 0.0 |

## AVERAGE PRECISION

Table 11: Mean average precision performance for dynamic link forecasting (window-based) over five runs for the continuous-time datasets. Values in parenthesis show the relative change as compared to the average precision performance for dynamic link prediction (batch-based).

| Dataset | JODIE | DyRep | TGN | TGAT | CAWN | EdgeBank | TCL | GraphMixer | DyGFormer | $\mu \pm \sigma$ |
|---|---|---|---|---|---|---|---|---|---|---|
| Enron | 80.8(↑12.0%) | 78.3(↑12.4%) | 68.6(↑5.0%) | 71.7(↑13.5%) | 76.0(↑15.1%) | 81.1(↑5.5%) | 78.2(↑11.3%) | 89.8(↑9.2%) | 85.3(↑11.6%) | 10.6%±3.4% |
| UCI | 87.0(↑7.2%) | 59.5(↑21.4%) | 69.2(↓2.8%) | 64.4(↓6.2%) | 64.0(↑1.6%) | 68.6(↑5.4%) | 65.2(↓5.5%) | 85.3(↓0.6%) | 80.5(↓0.2%) | 5.7%±6.4% |
| MOOC | 82.4(↓1.1%) | 76.9(↓0.3%) | 86.3(↓0.8%) | 82.7(↓2.1%) | 72.3(↓1.7%) | 59.1(↓2.7%) | 74.9(↓4.6%) | 74.4(↓4.5%) | 82.1(↓0.4%) | 2.0%±1.6% |
| Wiki. | 84.1(↓0.1%) | 80.9(↑0.1%) | 88.5(↓0.4%) | 87.5(↓0.4%) | 75.1(↑0.1%) | 73.3(↑0.2%) | 89.2(↓0.4%) | 90.8(↓0.4%) | 83.1(↑0.0%) | 0.2%±0.2% |
| LastFM | 76.7(↓1.1%) | 69.4(↓2.7%) | 78.8(↓1.9%) | 72.1(↓3.9%) | 68.2(↓2.3%) | 73.4(↑0.3%) | 70.3(↓1.8%) | 70.0(↓5.4%) | 80.5(↓0.7%) | 2.2%±1.6% |
| Myket | 64.5(↑1.6%) | 63.1(↑0.9%) | 62.8(↑1.2%) | 57.9(↑1.4%) | 46.6(↑3.3%) | 51.9(↑1.3%) | 58.9(↑1.1%) | 60.0(↑1.5%) | 46.1(↑3.3%) | 1.7%±0.9% |
| Social | 89.4(↑1.2%) | 91.9(↑0.3%) | 93.9(↑0.8%) | 95.0(↑0.2%) | 85.6(↓0.6%) | 79.7(↓1.1%) | 96.1(↓0.1%) | 95.8(↑0.4%) | 97.7(↑0.3%) | 0.6%±0.4% |
| Reddit | 80.1(↓0.0%) | 79.2(↑0.0%) | 80.6(↑0.0%) | 78.6(↓0.1%) | 81.3(↑0.2%) | 73.5(↓0.2%) | 76.5(↓0.0%) | 77.5(↓0.0%) | 82.8(↑0.1%) | 0.1%±0.1% |
| $\mu \pm \sigma$ | 3.0%±4.3% | 4.8%±7.9% | 1.6%±1.6% | 3.5%±4.6% | 3.1%±5.0% | 2.1%±2.2% | 3.1%±3.9% | 2.8%±3.3% | 2.1%±4.0% | |

Table 12: Mean average precision performance for dynamic link forecasting (window-based) over five runs for the discrete-time datasets. Values in parenthesis show the relative change as compared to the average precision performance for dynamic link prediction (batch-based).

| Dataset | JODIE | DyRep | TGN | TGAT | CAWN | EdgeBank | TCL | GraphMixer | DyGFormer | $\mu \pm \sigma$ |
|---|---|---|---|---|---|---|---|---|---|---|
| UN V. | 52.6(↓22.4%) | 49.6(↓26.8%) | 49.7(↓24.8%) | 52.7(↑0.9%) | 52.4(↑3.4%) | 84.2(↓0.7%) | 52.4(↓1.7%) | 54.0(↑0.1%) | 62.4(↑4.0%) | 9.4%±11.6% |
| US L. | 46.0(↓4.5%) | 62.5(↓14.5%) | 58.6(↓27.8%) | 71.0(↓0.3%) | 80.7(↓0.1%) | 63.2(↓0.2%) | 77.5(↑0.1%) | 86.5(↑0.6%) | 86.1(↑0.4%) | 5.4%±9.6% |
| UN Tr. | 52.7(↓10.6%) | 49.4(↓16.6%) | 53.2(↓9.7%) | 59.1(↑2.0%) | 59.2(↑2.4%) | 79.0(↓2.6%) | 57.5(↑2.3%) | 65.8(↑3.3%) | 67.1(↑4.4%) | 6.0%±5.1% |
| Can. P. | 52.1(↓1.3%) | 61.0(↓1.3%) | 69.9(↑2.0%) | 70.8(↑4.5%) | 68.3(↑6.9%) | 59.4(↓6.8%) | 68.2(↑6.2%) | 80.9(↑4.9%) | 83.2(↓14.3%) | 5.4%±4.0% |
| Flights | 65.2(↓2.2%) | 63.9(↓4.3%) | 68.3(↓0.0%) | 73.5(↑1.1%) | 64.7(↑1.3%) | 70.4(↓0.2%) | 71.0(↑0.4%) | 71.9(↑1.0%) | 68.9(↓0.1%) | 1.2%±1.4% |
| Cont. | 94.0(↓0.2%) | 95.8(↑0.4%) | 97.0(↑0.8%) | 96.8(↑0.8%) | 88.2(↑4.4%) | 89.4(↑0.6%) | 96.6(↑2.1%) | 95.7(↑1.6%) | 98.3(↑0.6%) | 1.3%±1.3% |
| $\mu \pm \sigma$ | 6.9%±8.5% | 10.6%±10.4% | 10.8%±12.5% | 1.6%±1.5% | 3.1%±2.4% | 1.8%±2.6% | 2.1%±2.2% | 1.9%±1.8% | 4.0%±5.4% | |

Table 13: Mean average precision performance over five runs for the test set of the continuous-time datasets from Poursafaei et al. (2022); Yu et al. (2023), including standard deviations.

| Eval | Dataset | JODIE | DyRep | TGN | TGAT | CAWN | EdgeBank | TCL | GraphMixer | DyGFormer |
|---|---|---|---|---|---|---|---|---|---|---|
| Forec. | Enron | 80.8 ± 5.3 | 78.3 ± 2.3 | 68.6 ± 5.6 | 71.7 ± 1.2 | 76.0 ± 0.7 | 81.1 ± 0.0 | 78.2 ± 2.9 | 89.8 ± 0.4 | 85.3 ± 0.6 |
| | UCI | 87.0 ± 1.9 | 59.5 ± 2.3 | 69.2 ± 1.0 | 64.4 ± 1.1 | 64.0 ± 0.7 | 68.6 ± 0.0 | 65.2 ± 0.9 | 85.3 ± 0.6 | 80.5 ± 0.9 |
| | MOOC | 82.4 ± 4.9 | 76.9 ± 4.3 | 86.3 ± 2.3 | 82.7 ± 0.7 | 72.3 ± 1.3 | 59.1 ± 0.0 | 74.9 ± 0.7 | 74.4 ± 0.6 | 82.1 ± 8.7 |
| | Wiki. | 84.1 ± 0.5 | 80.9 ± 0.4 | 88.5 ± 0.4 | 87.5 ± 0.2 | 75.1 ± 1.0 | 73.3 ± 0.0 | 89.2 ± 0.3 | 90.8 ± 0.2 | 83.1 ± 1.2 |
| | LastFM | 76.7 ± 0.6 | 69.4 ± 1.8 | 78.8 ± 3.5 | 72.1 ± 0.8 | 68.2 ± 0.5 | 73.4 ± 0.0 | 70.3 ± 6.5 | 70.0 ± 1.1 | 80.5 ± 0.9 |
| | Myket | 64.5 ± 1.8 | 63.1 ± 1.5 | 62.8 ± 2.2 | 57.9 ± 0.4 | 46.6 ± 0.2 | 51.9 ± 0.0 | 58.9 ± 2.6 | 60.0 ± 0.2 | 46.1 ± 1.7 |
| | Social | 89.4 ± 4.7 | 91.9 ± 1.0 | 93.9 ± 1.7 | 95.0 ± 0.3 | 85.6 ± 0.1 | 79.7 ± 0.0 | 96.1 ± 0.4 | 95.8 ± 0.2 | 97.7 ± 0.1 |
| | Reddit | 80.1 ± 0.3 | 79.2 ± 0.9 | 80.6 ± 0.6 | 78.6 ± 1.0 | 81.3 ± 0.4 | 73.5 ± 0.0 | 76.5 ± 0.6 | 77.5 ± 0.5 | 82.8 ± 0.8 |
| Pred. | Enron | 72.1 ± 3.0 | 69.7 ± 3.7 | 65.3 ± 3.2 | 63.2 ± 0.5 | 66.0 ± 0.5 | 76.9 ± 0.0 | 70.2 ± 3.4 | 82.3 ± 0.6 | 76.4 ± 0.4 |
| | UCI | 81.1 ± 3.3 | 49.0 ± 4.5 | 71.2 ± 1.1 | 68.6 ± 1.1 | 65.1 ± 0.6 | 65.0 ± 0.0 | 69.0 ± 0.8 | 85.9 ± 0.5 | 80.7 ± 1.1 |
| | MOOC | 83.4 ± 4.3 | 77.1 ± 3.8 | 87.0 ± 2.1 | 84.5 ± 0.7 | 73.5 ± 1.0 | 60.7 ± 0.0 | 78.5 ± 0.5 | 77.9 ± 0.8 | 82.4 ± 9.3 |
| | Wiki. | 84.1 ± 0.5 | 80.9 ± 0.3 | 88.8 ± 0.3 | 87.9 ± 0.2 | 75.0 ± 1.2 | 73.1 ± 0.0 | 89.5 ± 0.3 | 91.2 ± 0.2 | 83.1 ± 1.1 |
| | LastFM | 77.6 ± 0.6 | 71.4 ± 1.7 | 80.3 ± 3.2 | 75.0 ± 0.7 | 69.8 ± 0.5 | 73.2 ± 0.0 | 71.6 ± 6.1 | 74.1 ± 1.3 | 81.1 ± 0.9 |
| | Myket | 63.4 ± 1.7 | 62.5 ± 1.4 | 62.1 ± 2.3 | 57.1 ± 0.4 | 45.1 ± 0.2 | 51.3 ± 0.0 | 58.3 ± 2.2 | 59.1 ± 0.2 | 44.7 ± 1.6 |
| | Social | 88.3 ± 4.8 | 91.6 ± 0.7 | 93.2 ± 2.4 | 94.8 ± 0.3 | 86.2 ± 0.2 | 80.6 ± 0.0 | 96.2 ± 0.2 | 95.4 ± 0.1 | 97.3 ± 0.1 |
| | Reddit | 80.1 ± 0.3 | 79.2 ± 0.9 | 80.5 ± 0.5 | 78.6 ± 1.0 | 81.1 ± 0.4 | 73.7 ± 0.0 | 76.5 ± 0.6 | 77.5 ± 0.5 | 82.7 ± 0.8 |

Table 14: Mean average precision performance over five runs for the test set of the discrete-time datasets from Poursafaei et al. (2022); Yu et al. (2023), including standard deviations.

| Eval | Dataset | JODIE | DyRep | TGN | TGAT | CAWN | EdgeBank | TCL | GraphMixer | DyGFormer |
|---|---|---|---|---|---|---|---|---|---|---|
| Forec. | UN V. | 52.6 ± 1.8 | 49.6 ± 1.9 | 49.7 ± 3.9 | 52.7 ± 2.6 | 52.4 ± 2.0 | 84.2 ± 0.0 | 52.4 ± 0.9 | 54.0 ± 1.4 | 62.4 ± 1.7 |
| | US L. | 46.0 ± 0.9 | 62.5 ± 3.6 | 58.6 ± 2.4 | 71.0 ± 8.9 | 80.7 ± 3.7 | 63.2 ± 0.0 | 77.5 ± 4.3 | 86.5 ± 1.9 | 86.1 ± 1.0 |
| | UN Tr. | 52.7 ± 3.0 | 49.4 ± 0.9 | 53.2 ± 1.5 | 59.1 ± 2.7 | 59.2 ± 1.7 | 79.0 ± 0.0 | 57.5 ± 1.9 | 65.8 ± 1.9 | 67.1 ± 2.7 |
| | Can. P. | 52.1 ± 0.5 | 61.0 ± 7.6 | 69.9 ± 0.8 | 70.8 ± 1.6 | 68.3 ± 2.3 | 59.4 ± 0.0 | 68.2 ± 1.6 | 80.9 ± 0.5 | 83.2 ± 2.9 |
| | Flights | 65.2 ± 2.7 | 63.9 ± 2.8 | 68.3 ± 2.2 | 73.5 ± 0.3 | 64.7 ± 0.9 | 70.4 ± 0.0 | 71.0 ± 0.4 | 71.9 ± 0.8 | 68.9 ± 2.0 |
| | Cont. | 94.0 ± 2.6 | 95.8 ± 0.4 | 97.0 ± 0.5 | 96.8 ± 0.2 | 88.2 ± 0.2 | 89.4 ± 0.0 | 96.6 ± 0.4 | 95.7 ± 0.2 | 98.3 ± 0.0 |
| Pred. | UN V. | 67.8 ± 1.9 | 67.8 ± 1.7 | 66.1 ± 3.9 | 52.3 ± 2.5 | 50.7 ± 1.4 | 84.8 ± 0.0 | 53.3 ± 1.3 | 53.9 ± 1.7 | 60.0 ± 1.4 |
| | US L. | 48.2 ± 1.0 | 73.1 ± 2.2 | 81.2 ± 2.1 | 71.2 ± 8.2 | 80.8 ± 3.5 | 63.3 ± 0.0 | 77.4 ± 4.5 | 86.0 ± 2.0 | 85.8 ± 1.0 |
| | UN Tr. | 58.9 ± 3.1 | 59.3 ± 1.8 | 58.9 ± 1.5 | 57.9 ± 2.4 | 57.9 ± 2.1 | 81.1 ± 0.0 | 56.2 ± 1.5 | 63.8 ± 1.6 | 64.3 ± 2.2 |
| | Can. P. | 52.8 ± 0.5 | 61.8 ± 1.1 | 68.5 ± 2.1 | 67.7 ± 1.6 | 63.9 ± 1.3 | 63.8 ± 0.0 | 64.2 ± 2.0 | 77.1 ± 0.4 | 97.1 ± 0.7 |
| | Flights | 66.7 ± 3.3 | 66.8 ± 1.6 | 68.3 ± 1.8 | 72.7 ± 0.2 | 63.9 ± 0.9 | 70.5 ± 0.0 | 70.8 ± 0.5 | 71.2 ± 0.7 | 68.9 ± 1.8 |
| | Cont. | 94.2 ± 1.3 | 95.5 ± 0.3 | 96.3 ± 1.1 | 96.0 ± 0.3 | 84.5 ± 0.2 | 88.8 ± 0.0 | 94.6 ± 0.8 | 94.2 ± 0.1 | 97.7 ± 0.1 |

## F GLOBAL PERFORMANCE SCORES

The results presented in Table 3 and Table 4 assign the individual scores of each time window the same weight and then compute the mean over all scores to get the final score. This score measures the model performance across time, i.e. it is equally important for a model to perform well in periods that only have a few edge occurrences as well as in periods where many edges occur. In some scenarios, the focus might not be to forecast the existence of edges in all time windows equally well but instead forecast for all edges equally well. In the following, we investigate the model performance of link forecasting compared to link prediction using this perspective of model performance.

The results are presented in Table 15 for continuous-time temporal graphs and in Table 16 for discrete-time temporal graphs (corresponding average precision in Table 19 and Table 20). In contrast to the results presented in the main part of this work, the scores are computed over all edges instead of per time window and then averaged. As we can see, the changes between link forecasting and link prediction are less expressed if every edge is weighted the same instead of every time window. Nevertheless, we can still observe the patterns discussed above although not as distinct.

Table 15: Test AUC-ROC scores for link forecasting and the relative change compared to link prediction for continuous-time graphs on the *same trained models* (standard deviations in Table 17). We compute the AUC-ROC score over all edges instead of per time window or batch as in Table 3. The last row/column provides mean $\mu$ and standard deviation $\sigma$ of the absolute values of the relative change per column/row.

| Dataset | JODIE | DyRep | TGN | TGAT | CAWN | EdgeBank | TCL | GraphMixer | DyGFormer | $\mu \pm \sigma$ |
|---|---|---|---|---|---|---|---|---|---|---|
| Enron | 76.8(↑1.1%) | 73.4(↑0.2%) | 68.4(↑0.2%) | 58.9(↑0.2%) | 66.7(↑0.5%) | 78.5(↓1.6%) | 68.0(↑0.5%) | 81.1(↑0.8%) | 76.8(↑0.5%) | 0.6%±0.5% |
| UCI | 85.4(↑3.5%) | 60.7(↑18.1%) | 64.0(↑1.5%) | 59.5(↓0.2%) | 57.9(↑0.5%) | 71.3(↑3.1%) | 59.9(↓0.1%) | 80.5(↓0.1%) | 76.4(↑0.2%) | 3.0%±5.8% |
| MOOC | 83.9(↓0.9%) | 79.5(↓1.3%) | 88.1(↓0.5%) | 82.2(↑0.1%) | 70.5(↑0.2%) | 59.9(↓3.2%) | 72.3(↑0.0%) | 74.2(↑0.1%) | 81.0(↑0.0%) | 0.7%±1.0% |
| Wiki. | 81.6(↓0.2%) | 78.3(↓0.1%) | 84.1(↓0.0%) | 83.4(↓0.0%) | 71.6(↑0.1%) | 77.3(↑0.3%) | 85.1(↓0.0%) | 87.7(↓0.0%) | 80.2(↑0.2%) | 0.1%±0.1% |
| LastFM | 76.6(↓0.6%) | 70.2(↓1.4%) | 78.2(↓0.3%) | 68.5(↑0.0%) | 68.0(↓0.0%) | 78.0(↓0.2%) | 64.3(↓0.0%) | 66.1(↓0.0%) | 78.9(↓0.0%) | 0.3%±0.5% |
| Myket | 64.0(↑0.1%) | 64.0(↓0.0%) | 60.7(↓0.1%) | 57.4(↓0.3%) | 32.6(↑0.3%) | 52.0(↑0.0%) | 58.4(↓0.1%) | 59.4(↓0.1%) | 32.9(↑0.3%) | 0.1%±0.1% |
| Social | 90.4(↓0.6%) | 91.1(↓1.4%) | 91.5(↓0.1%) | 92.7(↑0.0%) | 87.8(↑0.1%) | 86.0(↑0.2%) | 95.3(↑0.1%) | 94.1(↑0.0%) | 97.5(↑0.0%) | 0.3%±0.5% |
| Reddit | 80.5(↓0.1%) | 79.5(↓0.1%) | 80.3(↓0.1%) | 78.6(↓0.1%) | 80.2(↓0.1%) | 78.5(↓0.2%) | 76.2(↓0.1%) | 77.1(↓0.1%) | 80.1(↓0.0%) | 0.1%±0.1% |
| $\mu \pm \sigma$ | 0.9%±1.1% | 2.8%±6.2% | 0.4%±0.5% | 0.1%±0.1% | 0.2%±0.2% | 1.1%±1.4% | 0.1%±0.2% | 0.1%±0.3% | 0.1%±0.2% | |

Table 16: Test AUC-ROC scores for discrete-time temporal graphs as in Table 15. All results with standard deviations are listed in Table 18.

| Dataset | JODIE | DyRep | TGN | TGAT | CAWN | EdgeBank | TCL | GraphMixer | DyGFormer | $\mu \pm \sigma$ |
|---|---|---|---|---|---|---|---|---|---|---|
| UN V. | 56.3(↓25.5%) | 53.3(↓28.7%) | 52.0(↓25.9%) | 54.3(↑2.8%) | 53.8(↑7.4%) | 89.7(↑0.1%) | 53.4(↑0.7%) | 57.1(↑1.4%) | 63.9(↑3.2%) | 10.6%±12.3% |
| US L. | 52.5(↓7.1%) | 61.8(↓22.1%) | 57.7(↓31.2%) | 78.6(↑0.1%) | 82.0(↑0.1%) | 68.4(↑1.3%) | 75.4(↓0.1%) | 90.4(↑0.3%) | 89.4(↑0.2%) | 6.9%±11.6% |
| UN Tr. | 57.6(↓13.1%) | 50.4(↓20.3%) | 54.4(↓14.0%) | 64.1(↑3.9%) | 67.6(↑4.6%) | 85.6(↓1.0%) | 63.7(↑4.5%) | 68.6(↑3.4%) | 70.7(↑3.5%) | 7.6%±6.5% |
| Can. P. | 64.0(↑0.5%) | 64.6(↓3.2%) | 72.7(↓1.4%) | 72.3(↑0.5%) | 68.1(↑0.0%) | 61.5(↓2.6%) | 68.3(↓0.1%) | 81.7(↑0.1%) | 83.7(↓14.3%) | 2.5%±4.6% |
| Flights | 67.3(↓3.0%) | 65.6(↓4.7%) | 68.1(↓1.0%) | 72.6(↓0.0%) | 65.2(↑0.3%) | 74.6(↓0.7%) | 70.5(↓0.1%) | 70.6(↓0.1%) | 68.5(↓0.5%) | 1.1%±1.7% |
| Cont. | 93.3(↓1.0%) | 94.1(↓1.4%) | 95.6(↓0.5%) | 95.3(↓0.0%) | 83.4(↑0.1%) | 92.2(0.0%) | 94.7(↑0.5%) | 93.7(↓0.0%) | 97.2(↓0.0%) | 0.4%±0.5% |
| $\mu \pm \sigma$ | 8.4%±9.6% | 13.4%±11.7% | 12.3%±13.6% | 1.2%±1.7% | 2.1%±3.2% | 0.8%±1.0% | 1.0%±1.7% | 0.9%±1.3% | 3.6%±5.5% | |

Table 17: Test AUC-ROC scores for link forecasting and link prediction averaged over 5 runs with standard deviations on continuous-time temporal graphs.

| Eval | Dataset | JODIE | DyRep | TGN | TGAT | CAWN | EdgeBank | TCL | GraphMixer | DyGFormer |
|------|---------|-------|-------|-----|------|------|----------|-----|------------|-----------|
| Forec. | Enron | 76.8 ± 3.9 | 73.4 ± 2.7 | 68.4 ± 3.4 | 58.9 ± 1.3 | 66.7 ± 0.5 | 78.5 ± 0.0 | 68.0 ± 5.7 | 81.1 ± 0.8 | 76.8 ± 0.5 |
| | UCI | 85.4 ± 1.0 | 60.7 ± 2.7 | 64.0 ± 1.1 | 59.5 ± 1.5 | 57.9 ± 0.6 | 71.3 ± 0.0 | 59.9 ± 0.8 | 80.5 ± 0.8 | 76.4 ± 0.5 |
| | MOOC | 83.9 ± 3.4 | 79.5 ± 4.1 | 88.1 ± 2.0 | 82.2 ± 0.6 | 70.5 ± 1.2 | 59.9 ± 0.0 | 72.3 ± 0.6 | 74.2 ± 1.4 | 81.0 ± 9.0 |
| | Wiki. | 81.6 ± 0.4 | 78.3 ± 0.4 | 84.1 ± 0.6 | 83.4 ± 0.2 | 71.6 ± 0.8 | 77.3 ± 0.0 | 85.1 ± 0.5 | 87.7 ± 0.3 | 80.2 ± 1.6 |
| | LastFM | 76.6 ± 0.5 | 70.2 ± 1.2 | 78.2 ± 3.0 | 68.5 ± 0.8 | 68.0 ± 0.3 | 78.0 ± 0.0 | 64.3 ± 6.0 | 66.1 ± 1.7 | 78.9 ± 0.6 |
| | Myket | 64.0 ± 2.1 | 64.0 ± 2.7 | 60.7 ± 2.3 | 57.4 ± 0.5 | 32.6 ± 0.4 | 52.0 ± 0.0 | 58.4 ± 2.0 | 59.4 ± 0.4 | 32.9 ± 1.0 |
| | Social | 90.4 ± 2.6 | 91.1 ± 1.0 | 91.5 ± 3.3 | 92.7 ± 0.5 | 87.8 ± 0.1 | 86.0 ± 0.0 | 95.3 ± 0.2 | 94.1 ± 0.2 | 97.5 ± 0.1 |
| | Reddit | 80.5 ± 0.2 | 79.5 ± 0.8 | 80.3 ± 0.4 | 78.6 ± 0.7 | 80.2 ± 0.3 | 78.5 ± 0.0 | 76.2 ± 0.4 | 77.1 ± 0.5 | 80.1 ± 1.1 |
| Pred. | Enron | 76.0 ± 3.0 | 73.2 ± 2.3 | 68.3 ± 2.9 | 58.8 ± 1.2 | 66.4 ± 0.4 | 79.8 ± 0.0 | 67.6 ± 5.6 | 80.5 ± 0.8 | 76.4 ± 0.5 |
| | UCI | 82.5 ± 1.3 | 51.4 ± 7.7 | 63.0 ± 1.3 | 59.6 ± 1.5 | 58.2 ± 0.6 | 69.1 ± 0.0 | 60.0 ± 0.9 | 80.7 ± 0.8 | 76.2 ± 0.6 |
| | MOOC | 84.6 ± 3.1 | 80.5 ± 3.2 | 88.5 ± 1.6 | 82.1 ± 0.6 | 70.3 ± 1.3 | 61.9 ± 0.0 | 72.3 ± 0.6 | 74.1 ± 1.4 | 81.0 ± 9.0 |
| | Wiki. | 81.7 ± 0.4 | 78.4 ± 0.4 | 84.1 ± 0.6 | 83.4 ± 0.2 | 71.5 ± 0.8 | 77.1 ± 0.0 | 85.2 ± 0.5 | 87.8 ± 0.3 | 80.0 ± 1.6 |
| | LastFM | 77.1 ± 0.7 | 71.2 ± 1.1 | 78.4 ± 2.7 | 68.5 ± 0.7 | 68.1 ± 0.3 | 78.2 ± 0.0 | 64.3 ± 6.0 | 66.1 ± 1.7 | 78.9 ± 0.6 |
| | Myket | 63.9 ± 2.1 | 64.0 ± 2.7 | 60.8 ± 2.3 | 57.5 ± 0.4 | 32.5 ± 0.4 | 51.9 ± 0.0 | 58.4 ± 2.0 | 59.5 ± 0.4 | 32.8 ± 1.0 |
| | Social | 91.0 ± 2.4 | 92.4 ± 0.4 | 91.5 ± 3.5 | 92.7 ± 0.5 | 87.7 ± 0.1 | 85.8 ± 0.0 | 95.2 ± 0.2 | 94.1 ± 0.2 | 97.4 ± 0.1 |
| | Reddit | 80.6 ± 0.1 | 79.5 ± 0.8 | 80.4 ± 0.4 | 78.7 ± 0.6 | 80.2 ± 0.3 | 78.6 ± 0.0 | 76.2 ± 0.4 | 77.1 ± 0.5 | 80.2 ± 1.1 |

Table 18: Test AUC-ROC scores for link forecasting and link prediction averaged over 5 runs with standard deviations on discrete-time temporal graphs.

| Eval | Dataset | JODIE | DyRep | TGN | TGAT | CAWN | EdgeBank | TCL | GraphMixer | DyGFormer |
|------|---------|-------|-------|-----|------|------|----------|-----|------------|-----------|
| Forec. | UN V. | 56.3 ± 1.4 | 53.3 ± 0.8 | 52.0 ± 7.2 | 54.3 ± 1.4 | 53.8 ± 2.1 | 89.7 ± 0.0 | 53.4 ± 1.1 | 57.1 ± 1.6 | 63.9 ± 1.7 |
| | US L. | 52.5 ± 1.8 | 61.8 ± 3.5 | 57.7 ± 1.8 | 78.6 ± 7.9 | 82.0 ± 4.0 | 68.4 ± 0.0 | 75.4 ± 5.3 | 90.4 ± 1.5 | 89.4 ± 0.9 |
| | UN Tr. | 57.6 ± 3.3 | 50.4 ± 1.2 | 54.4 ± 1.5 | 64.1 ± 1.3 | 67.6 ± 1.2 | 85.6 ± 0.0 | 63.7 ± 1.6 | 68.6 ± 2.6 | 70.7 ± 2.6 |
| | Can. P. | 64.0 ± 0.8 | 64.6 ± 7.5 | 72.7 ± 2.7 | 72.3 ± 2.6 | 68.1 ± 1.0 | 61.5 ± 0.0 | 68.3 ± 3.6 | 81.7 ± 0.9 | 83.7 ± 3.9 |
| | Flights | 67.3 ± 2.0 | 65.6 ± 1.8 | 68.1 ± 1.7 | 72.6 ± 0.2 | 65.2 ± 1.7 | 74.6 ± 0.0 | 70.5 ± 0.1 | 70.6 ± 0.3 | 68.5 ± 1.3 |
| | Cont. | 93.3 ± 1.9 | 94.1 ± 0.5 | 95.6 ± 0.5 | 95.3 ± 0.3 | 83.4 ± 0.1 | 92.2 ± 0.0 | 94.7 ± 0.5 | 93.7 ± 0.1 | 97.2 ± 0.0 |
| Pred. | UN V. | 75.6 ± 1.9 | 74.8 ± 1.2 | 70.2 ± 5.8 | 52.8 ± 1.6 | 50.1 ± 1.6 | 89.5 ± 0.0 | 53.0 ± 1.6 | 56.3 ± 2.0 | 61.9 ± 1.6 |
| | US L. | 56.5 ± 1.9 | 79.3 ± 1.0 | 84.0 ± 2.2 | 78.5 ± 7.8 | 81.9 ± 4.0 | 67.5 ± 0.0 | 75.4 ± 5.5 | 90.2 ± 1.5 | 89.3 ± 0.9 |
| | UN Tr. | 66.3 ± 3.0 | 63.2 ± 1.9 | 63.2 ± 1.2 | 61.7 ± 1.3 | 64.7 ± 1.3 | 86.4 ± 0.0 | 60.9 ± 1.3 | 66.3 ± 2.5 | 68.3 ± 2.3 |
| | Can. P. | 63.6 ± 0.7 | 66.8 ± 2.4 | 73.7 ± 3.5 | 72.0 ± 2.6 | 68.1 ± 1.0 | 63.1 ± 0.0 | 68.4 ± 3.6 | 81.6 ± 1.0 | 97.7 ± 0.6 |
| | Flights | 69.4 ± 2.3 | 68.9 ± 1.0 | 68.7 ± 1.6 | 72.6 ± 0.2 | 65.0 ± 1.4 | 74.6 ± 0.0 | 70.6 ± 0.1 | 70.7 ± 0.3 | 68.9 ± 1.1 |
| | Cont. | 94.3 ± 1.2 | 95.4 ± 0.3 | 96.1 ± 0.7 | 95.3 ± 0.3 | 83.3 ± 0.1 | 92.2 ± 0.0 | 94.3 ± 1.0 | 93.7 ± 0.1 | 97.3 ± 0.0 |

Table 19: Average Precision scores computed as in Table 15 for ROC-AUC scores on continuous-time temporal graphs. For a full list of results with standard deviations, see Table 21.

| Dataset | JODIE | DyRep | TGN | TGAT | CAWN | EdgeBank | TCL | GraphMixer | DyGFormer | $\mu \pm \sigma$ |
|---------|-------|-------|-----|------|------|----------|-----|------------|-----------|------------------|
| Enron | 69.8(↑1.1%) | 68.2(↓0.4%) | 65.3(↑0.5%) | 62.7(↑0.1%) | 66.8(↑0.4%) | 75.7(↓1.2%) | 70.7(↑0.9%) | 81.5(↑1.2%) | 77.1(↑0.8%) | 0.7%±0.4% |
| UCI | 85.1(↑6.8%) | 56.6(↑18.1%) | 71.9(↑0.4%) | 69.0(↓0.1%) | 65.6(↑0.0%) | 69.6(↓0.1%) | 85.9(↓0.2%) | 81.2(↑0.2%) | 3.2%±6.0% |
| MOOC | 82.8(↓0.5%) | 76.2(↓0.6%) | 86.3(↓0.7%) | 84.4(↑0.0%) | 73.6(↑0.1%) | 58.7(↓3.2%) | 78.3(↓0.0%) | 77.7(↓0.0%) | 82.1(↑0.0%) | 0.6%±1.0% |
| Wiki. | 84.1(↑0.0%) | 80.8(↑0.2%) | 88.8(↓0.0%) | 87.9(↑0.0%) | 75.2(↑0.3%) | 73.4(↑0.5%) | 89.5(↓0.0%) | 91.1(0.0%) | 83.3(↓0.3%) | 0.1%±0.2% |
| LastFM | 76.4(↓1.9%) | 70.3(↓2.5%) | 78.5(↓0.7%) | 76.0(↑0.0%) | 72.2(↓0.0%) | 73.5(↓0.0%) | 72.5(↑0.0%) | 75.1(↓0.0%) | 82.1(↑0.0%) | 0.6%±1.0% |
| Myket | 63.1(↑0.4%) | 61.7(↓0.0%) | 61.3(↑0.0%) | 56.4(↓0.3%) | 44.8(↑0.0%) | 51.1(↑0.0%) | 57.6(↓0.1%) | 58.7(↓0.1%) | 44.4(↑0.0%) | 0.1%±0.2% |
| Social | 87.0(↓1.0%) | 90.0(↓1.5%) | 93.0(↑0.0%) | 94.9(↑0.0%) | 86.6(↑0.2%) | 80.8(↑0.3%) | 96.4(↑0.1%) | 95.5(↑0.0%) | 97.6(↑0.1%) | 0.4%±0.5% |
| Reddit | 79.7(↓0.1%) | 78.9(0.0%) | 80.3(↓0.1%) | 78.3(↓0.1%) | 81.0(↑0.1%) | 73.4(↓0.2%) | 76.3(↓0.0%) | 77.2(↓0.0%) | 82.6(↓0.0%) | 0.1%±0.1% |
| $\mu \pm \sigma$ | 1.5%±2.2% | 2.9%±6.2% | 0.3%±0.3% | 0.1%±0.1% | 0.1%±0.2% | 1.1%±1.3% | 0.1%±0.3% | 0.2%±0.4% | 0.2%±0.3% | |

Table 20: Average Precision scores as in Table 19 for discrete-time graphs. All results and standard deviations are listed in Table 22.

| Dataset | JODIE | DyRep | TGN | TGAT | CAWN | EdgeBank | TCL | GraphMixer | DyGFormer | $\mu \pm \sigma$ |
|---------|-------|-------|-----|------|------|----------|-----|------------|-----------|------------------|
| UN V. | 53.3(↓23.6%) | 50.6(↓28.2%) | 49.9(↓22.6%) | 52.5(↑1.7%) | 52.6(↑4.8%) | 84.1(↓0.4%) | 52.4(↓0.9%) | 54.1(↑1.4%) | 60.0(↑3.0%) | 9.6%±11.6% |
| US L. | 46.0(↓4.2%) | 62.5(↓15.3%) | 58.6(↓28.5%) | 71.0(↑0.2%) | 80.7(↑0.0%) | 63.2(↓0.0%) | 77.5(↓0.1%) | 86.5(↑0.7%) | 86.1(↑0.6%) | 5.5%±9.9% |
| UN Tr. | 52.8(↓10.0%) | 49.6(↓15.7%) | 53.3(↓9.6%) | 59.1(↑3.0%) | 59.2(↑3.3%) | 79.0(↓2.5%) | 57.5(↑3.5%) | 65.8(↑2.7%) | 67.1(↑3.3%) | 6.0%±4.7% |
| Can. P. | 52.3(↑0.4%) | 59.9(↓6.4%) | 69.7(↓2.5%) | 70.5(↑0.1%) | 66.6(↑0.1%) | 58.0(↓2.8%) | 67.0(↑0.1%) | 81.4(↑0.2%) | 82.2(↓16.1%) | 3.2%±5.3% |
| Flights | 65.2(↓2.3%) | 63.4(↓5.3%) | 68.3(↓0.8%) | 73.5(↓0.0%) | 64.7(↑0.7%) | 70.3(0.0%) | 71.0(↓0.1%) | 71.9(↓0.0%) | 68.8(↓0.6%) | 1.1%±1.7% |
| Cont. | 90.2(↓2.2%) | 95.1(↓0.7%) | 95.7(↓0.7%) | 96.0(↓0.0%) | 85.2(↓0.0%) | 88.7(↓0.1%) | 95.4(↑0.4%) | 93.5(↓0.1%) | 97.9(↓0.0%) | 0.5%±0.7% |
| $\mu \pm \sigma$ | 7.1%±8.7% | 11.9%±9.9% | 10.8%±12.0% | 0.8%±1.2% | 1.5%±2.1% | 1.0%±1.3% | 0.8%±1.3% | 0.8%±1.0% | 3.9%±6.1% | |

Table 21: Test average precision scores for link forecasting and link prediction averaged over 5 runs with standard deviations on continuous-time temporal graphs.

| Eval | Dataset | JODIE | DyRep | TGN | TGAT | CAWN | EdgeBank | TCL | GraphMixer | DyGFormer |
|---|---|---|---|---|---|---|---|---|---|---|
| Forec. | Enron | 69.8 ± 3.6 | 68.2 ± 4.0 | 65.3 ± 2.8 | 62.7 ± 0.8 | 66.8 ± 0.5 | 75.7 ± 0.0 | 70.7 ± 3.7 | 81.5 ± 0.6 | 77.1 ± 0.7 |
| | UCI | 85.1 ± 1.8 | 56.6 ± 2.4 | 71.9 ± 0.9 | 69.0 ± 1.0 | 65.6 ± 0.7 | 66.9 ± 0.0 | 69.6 ± 0.8 | 85.9 ± 0.4 | 81.2 ± 0.9 |
| | MOOC | 82.8 ± 5.1 | 76.2 ± 4.3 | 86.3 ± 2.7 | 84.4 ± 0.7 | 73.6 ± 0.9 | 58.7 ± 0.0 | 78.3 ± 0.6 | 77.7 ± 0.7 | 82.1 ± 9.8 |
| | Wiki. | 84.1 ± 0.5 | 80.8 ± 0.3 | 88.8 ± 0.4 | 87.9 ± 0.2 | 75.2 ± 1.1 | 73.4 ± 0.0 | 89.5 ± 0.3 | 91.1 ± 0.2 | 83.3 ± 1.2 |
| | LastFM | 76.4 ± 0.5 | 70.3 ± 2.0 | 78.5 ± 3.9 | 76.0 ± 0.7 | 72.2 ± 0.4 | 73.0 ± 0.0 | 72.5 ± 5.9 | 75.1 ± 1.2 | 82.1 ± 0.8 |
| | Myket | 63.1 ± 1.8 | 61.7 ± 1.5 | 61.3 ± 2.1 | 56.4 ± 0.4 | 44.8 ± 0.3 | 51.1 ± 0.0 | 57.6 ± 2.3 | 58.7 ± 0.2 | 44.4 ± 1.7 |
| | Social | 87.0 ± 6.1 | 90.0 ± 1.4 | 93.0 ± 2.4 | 94.9 ± 0.3 | 86.6 ± 0.1 | 80.8 ± 0.0 | 96.4 ± 0.2 | 95.5 ± 0.2 | 97.6 ± 0.1 |
| | Reddit | 79.7 ± 0.4 | 78.9 ± 0.9 | 80.3 ± 0.6 | 78.3 ± 1.0 | 81.0 ± 0.5 | 73.4 ± 0.0 | 76.3 ± 0.6 | 77.2 ± 0.5 | 82.6 ± 0.8 |
| Pred. | Enron | 69.0 ± 2.1 | 68.5 ± 4.3 | 65.0 ± 3.9 | 62.6 ± 0.6 | 66.6 ± 0.5 | 76.7 ± 0.0 | 70.1 ± 3.6 | 80.5 ± 0.6 | 76.5 ± 0.6 |
| | UCI | 79.7 ± 3.0 | 48.0 ± 4.3 | 71.7 ± 1.1 | 69.1 ± 1.0 | 65.6 ± 0.7 | 64.9 ± 0.0 | 69.6 ± 0.8 | 86.0 ± 0.5 | 81.0 ± 1.0 |
| | MOOC | 83.2 ± 4.5 | 76.6 ± 4.0 | 86.9 ± 2.2 | 84.3 ± 0.7 | 73.5 ± 1.0 | 60.6 ± 0.0 | 78.3 ± 0.5 | 77.7 ± 0.7 | 82.1 ± 9.8 |
| | Wiki. | 84.1 ± 0.6 | 80.6 ± 0.3 | 88.8 ± 0.3 | 87.9 ± 0.2 | 75.0 ± 1.1 | 73.0 ± 0.0 | 89.5 ± 0.3 | 91.1 ± 0.2 | 83.1 ± 1.2 |
| | LastFM | 77.9 ± 0.7 | 72.1 ± 1.9 | 79.1 ± 3.1 | 76.0 ± 0.6 | 72.2 ± 0.4 | 73.1 ± 0.0 | 72.4 ± 5.9 | 75.1 ± 1.2 | 82.1 ± 0.8 |
| | Myket | 62.9 ± 1.8 | 61.7 ± 1.5 | 61.3 ± 2.1 | 56.6 ± 0.4 | 44.8 ± 0.3 | 51.1 ± 0.0 | 57.6 ± 2.2 | 58.7 ± 0.2 | 44.4 ± 1.7 |
| | Social | 87.8 ± 5.3 | 91.3 ± 0.8 | 93.0 ± 2.6 | 94.9 ± 0.3 | 86.4 ± 0.1 | 80.5 ± 0.0 | 96.3 ± 0.2 | 95.4 ± 0.1 | 97.6 ± 0.1 |
| | Reddit | 79.8 ± 0.4 | 78.9 ± 0.9 | 80.4 ± 0.6 | 78.4 ± 1.0 | 80.9 ± 0.5 | 73.6 ± 0.0 | 76.3 ± 0.6 | 77.2 ± 0.5 | 82.6 ± 0.8 |

Table 22: Test average precision scores for link forecasting and link prediction averaged over 5 runs with standard deviations on discrete-time temporal graphs.

| Eval | Dataset | JODIE | DyRep | TGN | TGAT | CAWN | EdgeBank | TCL | GraphMixer | DyGFormer |
|---|---|---|---|---|---|---|---|---|---|---|
| Forec. | UN V. | 53.3 ± 1.2 | 50.6 ± 1.5 | 49.9 ± 4.4 | 52.5 ± 1.4 | 52.6 ± 1.9 | 84.1 ± 0.0 | 52.4 ± 0.9 | 54.1 ± 1.4 | 60.0 ± 2.0 |
| | US L. | 46.0 ± 0.9 | 62.5 ± 3.6 | 58.6 ± 2.4 | 71.0 ± 8.9 | 80.7 ± 3.7 | 63.2 ± 0.0 | 77.5 ± 4.3 | 86.5 ± 1.9 | 86.1 ± 1.0 |
| | UN Tr. | 52.8 ± 3.1 | 49.6 ± 0.8 | 53.3 ± 1.7 | 59.1 ± 2.7 | 59.2 ± 1.7 | 79.0 ± 0.0 | 57.5 ± 1.9 | 65.8 ± 1.9 | 67.1 ± 2.7 |
| | Can. P. | 52.3 ± 0.6 | 59.9 ± 6.5 | 69.7 ± 1.5 | 70.5 ± 1.8 | 66.6 ± 2.1 | 58.0 ± 0.0 | 67.0 ± 1.9 | 81.4 ± 0.5 | 82.2 ± 3.2 |
| | Flights | 65.2 ± 2.6 | 63.4 ± 2.6 | 68.3 ± 2.2 | 73.5 ± 0.3 | 64.7 ± 0.8 | 70.3 ± 0.0 | 71.0 ± 0.4 | 71.9 ± 0.8 | 68.8 ± 2.0 |
| | Cont. | 90.2 ± 5.2 | 95.1 ± 0.6 | 95.7 ± 1.0 | 96.0 ± 0.4 | 85.2 ± 0.2 | 88.7 ± 0.0 | 95.4 ± 0.6 | 93.5 ± 0.1 | 97.9 ± 0.1 |
| Pred. | UN V. | 69.7 ± 1.5 | 70.5 ± 1.1 | 64.4 ± 6.5 | 51.6 ± 1.3 | 50.2 ± 1.4 | 84.5 ± 0.0 | 52.9 ± 1.4 | 53.3 ± 1.7 | 58.2 ± 1.5 |
| | US L. | 48.0 ± 1.0 | 73.8 ± 2.4 | 81.9 ± 2.3 | 70.9 ± 8.7 | 80.7 ± 3.6 | 63.2 ± 0.0 | 77.6 ± 4.4 | 85.8 ± 2.0 | 85.6 ± 1.2 |
| | UN Tr. | 58.7 ± 3.7 | 58.9 ± 1.7 | 58.9 ± 1.5 | 57.4 ± 2.6 | 57.4 ± 2.3 | 81.0 ± 0.0 | 55.6 ± 1.5 | 64.1 ± 1.7 | 64.9 ± 2.6 |
| | Can. P. | 52.1 ± 0.4 | 64.0 ± 1.7 | 71.5 ± 1.8 | 70.4 ± 1.7 | 66.5 ± 2.4 | 59.7 ± 0.0 | 67.0 ± 1.9 | 81.2 ± 0.4 | 98.0 ± 0.5 |
| | Flights | 66.7 ± 3.6 | 66.9 ± 1.9 | 68.9 ± 2.0 | 73.5 ± 0.3 | 64.2 ± 1.0 | 70.3 ± 0.0 | 71.0 ± 0.4 | 71.9 ± 0.8 | 69.3 ± 2.0 |
| | Cont. | 92.2 ± 2.3 | 95.8 ± 0.4 | 96.4 ± 0.9 | 96.0 ± 0.4 | 85.2 ± 0.2 | 88.7 ± 0.0 | 95.0 ± 1.0 | 93.5 ± 0.1 | 98.0 ± 0.0 |

## G  NORMALIZED MUTUAL INFORMATION

Normalized mutual information is an information-theoretic measure based on mutual information. It is based on mutual information, which for two random variables $X$ and $Y$ captures the bits of information we gain about the outcome of $Y$ if we know the outcome of $X$ and vice-versa. A formal definition is given in the following:

**Definition** (Mutual Information)**.** Consider two random variables, $X$ and $Y$ with joint probability mass function $p(x, y)$ and marginal probability mass functions $p(x)$ and $p(y)$. Mutual information is the reduction in the uncertainty of $X$ due to the knowledge of $Y$ defined as (Cover & Thomas, 2006)

$$I(X, Y) = \sum_{x \in X} \sum_{y \in Y} p(x, y) \log \frac{p(x, y)}{p(x) p(y)}.$$

Note that mutual information can be defined using different logarithms. The intuitive understanding described above using bits of information is defined using $\log_2$. This work utilizes the implementation of the Python library scikit-learn (Pedregosa et al., 2011) which uses the natural logarithm $\log_e$.

The specific value of mutual information depends on the entropy

$$H(X) = - \sum_{x \in X} p(x) \log p(x)$$

of the underlying random variables, and is thus difficult to compare across different settings. To address this issue, normalized mutual information provides a measure between zero and one that is normalized based on the entropies of the underlying random variables (Vinh et al., 2010). We use the following normalization as implemented in scikit-learn's function `normalized_mutual_info_score`:

$$\text{NMI}(X, Y) = \frac{I(X, Y)}{\frac{1}{2} \cdot (H(X) + H(Y))}$$

In the context of our work, we use the NMI to capture the loss of temporal information. In Figure 4, we use the NMI to measure how much information about the edges' timestamps is lost by grouping the edges into batches. Specifically, we use the number $i$ of each batch $B_i^+ \cup B_i^-$ of the definition of dynamic link prediction in Section 2 assigned to each edge in the corresponding batch as one random variable and the timestamps of the edges as the other. With this setup, we can measure how much information about the timestamps of edges we gain – or keep – based on the batch number only.

Additionally, we use the NMI in Table 2 to quantify the difference between the assignments of edges to time windows and batches respectively. Similar as above, we assign to each edge in the test set of each dataset its batch number and also a time window number corresponding to the time window the edge belongs to and then compute the NMI between those two variables.

