# OpenReview forum: "From Link Prediction to Forecasting: Information Loss in Batch-based Temporal Graph Learning"
_ICLR.cc/2025/Conference — Submitted to ICLR 2025_

### Official Review · Reviewer_sDdG · 2024-11-01

**Soundness:** 3
**Presentation:** 3
**Contribution:** 2
**Rating:** 5
**Confidence:** 5

**Summary:**

This work tackles the issue of information loss and leakage in the batch-oriented evaluation protocol for temporal graph learning. It first validates the existence of such loss and leakage through experiments that compute the NMI between batches and timestamps. To address this, the authors propose a new evaluation protocol for link forecasting and re-evaluate numerous existing methods within this framework. The provided code in the supplementary material appears comprehensive and sufficient.

**Strengths:**

The paper presents a study that provides a novel perspective on the internal issues with the batch-oriented evaluation protocol. The overall presentation is clear, and it includes adequate background information. The experiments related to the new evaluation protocol are relatively comprehensive, with re-evaluation of numerous existing methods. However, there are some missing aspects (as noted in the identified weaknesses) that should be addressed.

**Weaknesses:**

W1. Figure 2 and Figure 3 offer trivial or intuitive results, which seem to be redundant in the main text. I would suggest to just put part of them in the main text and the remaining in the appendix. Otherwise, these two figures would attract too much attention from the readers and make them lost before reading your point (Figure 4).

W2. The NMI may not be a trivial concept for common readers, therefore it would be helpful to include some technical details of it in the appendix and link to it at line 263. The current presentation of this paragraph is quite confusing as I cannot understand what lines 266-268 mean (how the NMI was computed). Maybe some formulation would help improve the clarification here.

W3. While the issue with the batch-oriented evaluation protocol is effectively identified using NMI, the paper does not adequately explain how this impacts previous evaluations and benchmarks that use this protocol. For instance, could you provide experimental validation to demonstrate the biases or flaws in past evaluations? The experiments in the current manuscript are associated with the new link forecasting protocol. Still, I am confused about how to experimentally validate that this protocol is better than the previous ones.

W4. One advantage of the batch-oriented evaluation is the efficiency, but there seems to be no comparison or comment regarding the training/evaluation efficiency of the link forecasting protocol. I am aware of line 393, Computational cost, but I think such a not in-depth analysis is not enough. Some experimental results could be included.

W5. This is a relatively minor point, as is included in the limitation. For the frameworks that consider the temporal link prediction problem as a ranking task, is it possible to have more discussions about the pros/cons of time-window-based approaches and ranking approaches?

Minor:
1. line 143, should be {$t_{b\cdot i}\, ..., t_{b\cdot (i+1)-1}$}? Also, line 244 seems to have the same mistake.
2. line 146, 147. if there is no constraint on (u,v), then it's possible that (u,v) is not in the B+ or B- batch. Is something like $(u, v) \in B^+ \cup B^-$ missing? I think similar issue exists for you link forecasting definition, line 379.
3. line 346. "negative edges that do not occur in time window [i · h,(i + 1) · h)". I feel it ambiguous. is those negative edges not in this time window, or in this time window but do not occur (interact)? Should be the latter?
4. Actually it could be an independent section for related work. I would recommend commenting on how existing TGNN benchmarks flow to validate that the widely-used batch-oriented training/evaluation does have issues that this work is trying to fix.
5. The code implementation largely builds upon an existing framework, DyGLib, and it would be helpful to include comments or documentation clarifying where the extensions and modifications occur at the code level (code snippets) in the appendix.

**Questions:**

I currently lean towards a rejection of the manuscript but am very open to further discussion and increasing the based on how well my concerns get addressed. I encourage the authors to refer to the identified weaknesses.

---

> ### Author Response · Authors · 2024-11-21
>
> Thank you for your detailed review and for highlighting that our paper "provides a novel perspective on the internal issues with the batch-oriented evaluation protocol".
> We respond to your identified points for improvement in the following:
>
> ### W1
>
> We respectfully disagree that the results in Figure 2 (histogram of edge activities) and 3 (batch durations for different batch sizes) are trivial. While the results are indeed intuitive to understand, we still believe that investigating the characteristics of temporal graphs shown in Figures 2 and 3 is crucial to highlight the potential biases and pitfalls in batch-based evaluations, which is why we included them.
> If the reviewer strongly disagrees with this point, we could possibly move Figure 3 to the appendix but would prefer to keep it in the main text.
>
> ### W2
>
> Thank you for pointing this out, we have added a more detailed explanation in the form of additional Appendix G to the uploaded revision of our paper.
>
> ### W3
>
> We thank the reviewer for these crucial questions, which we are happy to clarify below.
> The goal of our work is to facilitate a more realistic evaluation and a fairer comparison between different models by fixing problems with the prediction task. What do we mean by this? The current formulation of dynamic link prediction allows tuning the batch size as an implicit hyperparameter that - as we show in our work - critically affects model performance. But, as we also show in our work, changing the batch size fundamentally changes the characteristics of the prediction problem: the resulting batches span different lengths and the NMI between edges' timestamps and their batch numbers change. Moreover, we argue that the real-world characteristics of a link prediction setting suggest a sensible time horizon (we suggest horizons for the used datasets in Appendix C). In that sense, we argue that the task definition provided in our work (dynamic link forecasting) is "better" as it more closely reflects reality and is thus consistent across different models.
>
> Moreover, our experiments uncover a previously unseen bias toward memory-based models due to the information leakage that we identified in our work. Using a batch-based training pipeline and then evaluating the trained models on both the batch-based and our time-window-based approach, we observe a substantial drop in performance for memory-based models using our evaluation approach compared to the batch-based strategy. This drop in performance validates the identified information leakage that is fixed using our evaluation strategy.
>
> ### W4
>
> Thank you for this idea. We agree that reporting the runtime of our time-window-based approach and the runtime of the batch-based approach will improve our paper. Sadly, we did not record the runtime during our initial experiments and it is not feasible to rerun all experiments during the rebuttal. However, we propose to include them in the camera-ready version.
>
> ### W5
>
> We can see that our explanation in the limitations section (lines 528-529) is rather vague and we are happy to provide more details in the uploaded revision as well as below.
> Ranking-based approaches typically sample a large number (e.g. 100 for "tgbl-review" [1]) of negative samples for each positive sample.
> Commonly, this combination of one positive and many negative samples for this positive edge is then used as a batch.
> While the problem of information leakage remains because there might still be multiple batches with the same timestamp, using only one positive sample per batch alleviates the information loss since each batch only comprises the single timestamp of the positive edge.
> This leads to a better estimation of the models' precision but also increases the runtime of the evaluation by a large amount.
> Our time-window-based approach is much faster because it uses fewer negative samples.
> The model's precision, on the other hand, may not be as accurate.
> However, we argue that this is a reasonable simplification since in real-world scenarios the temporal resolution of the available data is often not the same as the one that is necessary for the prediction, e.g. while data on past purchases for each customer might be available in a per-second resolution it is enough to provide customer recommendations every hour or day.

---

> > ### Author Response · Authors · 2024-11-21
> >
> > ### Minor
> >
> > 1. Yes, this is a typo. Thank you for catching it. We fixed it in the uploaded revision.
> > 2. Thank you for the suggestion. This, indeed, improves the clarity of the definitions and is included in the uploaded revision.
> > 3. Thank you for this question. We added a clarification in the uploaded revision. The sentence is now as follows: We use $W_i^-$ to denote a sample of $|W_i^+|$ negative edges that are sampled using one of the negative sampling approaches described in \cref{app:negative_sampling} and do not occur as positive edges in time window $[i \cdot h,(i+1) \cdot h)$, i.e.\ $W_i^- \cap W_i^+ = \emptyset$.
> > 4. Thank you for this interesting idea. We are very much in favor of including such an appendix, especially since some of the problems we address are rather technical and depend on the specific implementation details of each method or training strategy. However, going through the implementations of all related work is not feasible in the remaining time of the rebuttal. We, thus, propose to include such an appendix in the camera-ready version.
> > 5. We respectfully disagree that explanations about the specific implementation that we used should be included in the paper. We argue that the paper itself should be formulated in a general way that makes it possible to implement our approach in any available software framework. We will, however, release a de-anonymized GitHub repository with descriptions and documentation that make it easy to see our modifications compared to the framework our code is based on and provide the link in the camera-ready version.
> >
> > ### References
> >
> > [1] Shenyang Huang et al. Temporal Graph Benchmark for Machine Learning on Temporal Graphs. In NeurIPS, 2023.

---

> > > ### Comment · Reviewer_sDdG · 2024-11-28
> > > **Thanks for clarifications on the minors**
> > >
> > > Thank you for your clarifications on the minor weaknesses. In my opinion, most of these can be easily addressed and do not impact my overall evaluation of the paper. Regarding point 5, specifically whether to explicitly discuss the implementation in the paper, I have seen both approaches—some papers include detailed implementation discussions, while others do not. I appreciate that the authors have provided the source code. Highlighting the modifications either within the paper or in the README/documentation of the source code would both be effective ways to help readers better understand the implementation. I believe the authors are free to choose the approach that best fits their work.

---

> > ### Comment · Reviewer_sDdG · 2024-11-28
> > **Thanks for your clarifications**
> >
> > >W1 Figure 2 and 3
> >
> > This is a relatively minor point and does not significantly impact my overall evaluation of the manuscript. While the authors are free to decide what to include in the main text, I suggest that it may not be necessary to display all datasets there. Instead, selecting one or two representative examples for the main text and moving the rest to the appendix could improve readability and focus.
> >
> > > W2 NMI explanation
> >
> > Thanks for having an appendix section to explain what NMI is for common readers I think it is pretty clear now.
> >
> > > W3 Comparison with previous evaluation protocol.
> >
> > I appreciate the clarity of the overall story and argument presented in this manuscript. Your arguments intuitively make sense to me, but I wonder if they could be further strengthened with some experimental evidence to validate the intuition scientifically. For instance, a case study of several algorithms could demonstrate how they exploit or "hack" the previous protocol. Evidence might include intermediate results (e.g., values prior to the final AUC-ROC) showing discrepancies or abnormalities, or unusual observations made during evaluations using the previous protocol—potentially highlighting how your research idea for this paper was identified.
> >
> > > W4 Efficiency
> >
> > I understand that the rebuttal period is significantly shorter than the time it took to complete this research. However, I would appreciate it if the authors could provide some sample results—there’s no need to report all results, but sharing a subset would help clarify and strengthen the argument of efficiency.
> >
> > > W5 pros/cons of time-window-based approaches and ranking approaches
> >
> > Thank you for your clarifications. Please ensure these points are integrated into the camera-ready version.
> >
> > The rebuttal effectively addresses some of my concerns (W1, W2, and W5). However, the remaining issues (W3 and W4) could benefit from further elaboration, with W3 being the most critical in my view. At this stage, I am inclined to increase my score to 5, but I will make a final decision at the end of the rebuttal period.

---

> > > ### Author Response · Authors · 2024-12-02
> > >
> > > Thank you for your response and your clarification regarding W3, as we apparently misunderstood your comment. Following your suggestion, we re-examined our experimental results to find empirical evidence that supports our claims. We report our findings in the following as well as the runtime of both evaluation approaches on some of the datasets.
> > >
> > > ### W3
> > >
> > > The results reported in Tables 3 and 4 show average AUC-ROC scores for each batch or window, which is the common way to report results when evaluating dynamic link prediction and the way it is implemented in DyGLib. To provide insights into the reasons behind the performance changes between both approaches, instead of taking the average, we visualize the AUC-ROC for each individual batch and show the AUC-ROC of each model for the discrete-time UN Trade dataset in the following:
> > >
> > > https://ibb.co/w6vFf8g
> > >
> > > We observe that for batches of the same snapshot (snapshot borders are visualized with dashed grey vertical lines) the AUC-ROC increases significantly (the null hypothesis that there is no linear relationship is rejected with $p<0.05$; for JODIE, DyRep and TGN the $p$-values are smaller than or equal to $0.005$ on all snapshots) with increasing batch number for memory-based models. For non-memory-based models, we either observe no significant changes (TGAT, CAWN, and TCL) or a decreasing AUC-ROC with increasing batch number within the same snapshot (GraphMixer and DyGFormer). For discrete-time datasets, this supports our claim that there is an information leakage that gives memory-based models an unfair advantage which allows these models to use information from timestamp $t$ to predict edges at timestamp $t$.
> > > This highlights that the drop in performance on discrete-time datasets for memory-based models when comparing our window-based to the batch-based approach is because our approach mitigates the issue of information leakage.
> > >
> > > For continuous-time datasets, a major issue with batch-based evaluation is that the nature of the dynamic link prediction task changes depending on the activity distribution in a temporal graph. In the following plot, we show the distribution of time stamps across all batches and time windows (with batch size 200 and the according time window duration as selected in Section 4) for a batch- and window-based evaluation in the Enron data set:
> > >
> > > https://ibb.co/GFmx4g7
> > >
> > > For an evaluation with a fixed batch size, 94% of the batches occur during the first half of the observation time. The second half of the observation time with less activity is aggregated in a very small number of batches. Our time-window approach is not affected by this decrease in activity because time windows have a fixed time period rather than a fixed amount of edges.
> > >
> > > To showcase how this affects the performance of different models we compare the AUC-ROC values of the window- and batch-based approach across time for the Enron data set. The x-axis shows the average timestamp of each batch/window:
> > >
> > > https://ibb.co/VL0r0hR
> > >
> > > During the second half of the observation period, our approach leads to more time windows compared to the number of batches. For our window-based approach, most models perform better during this period compared to their performance in the first half. This increase in performance during the second half of the observation period is masked by the batch-based evaluation, which aggregates this long period of time in a very small number of batches.
> > >
> > > The plot further highlights that the batch-based evaluation is affected by an anomaly at hour 28356, where the performance of all models drops substantially. At this time, one employee sent 1705 emails at once, resulting in 1705 temporal edges with identical time stamps that are evaluated in 9 consecutive batches (which are again subject to the information leakage issue discussed above). In contrast, our approach only uses a single time window for these 1705 edges with identical time stamps.
> > >
> > > In summary, we believe that these further insights substantiate our claims and we will be happy to include them in the camera-ready version (along with the same analysis for the other data sets).

---

> > > ### Author Response · Authors · 2024-12-02
> > >
> > > ### W4
> > >
> > > The following table shows the average runtime of both evaluation approaches (batch-based and window-based) across five runs in seconds. Note that some cells currently contain NaN-values and not all datasets are included because the experiments for these configurations are not yet finished. For most continuous-time datasets and models, the table shows that both approaches have a comparable runtime, supporting our claims that our approach is comparable in runtime efficiency when the windows contain, on average, a similar number of edges compared to a batch. Note that due to the inhomogeneously distributed temporal activity of the UCI dataset and the chosen time window duration (to get an average of 200 edges per window), the time windows in the test set generally contain less than 200 edges resulting in a longer runtime for the window-based approach than the batch-based approach. Our approach is faster for discrete-time datasets since, instead of using a fixed number of 200 edges per batch, we use each snapshot as a whole. The snapshots typically contain more than 200 edges, leading to more edges being processed in parallel and, thus, a faster evaluation.
> > >
> > > | Dataset   | Approach   | JODIE | DyRep | TGN | TGAT | CAWN | TCL | GraphMixer | DyGFormer |
> > > |:---|:---|:----|:-----|:----|:----|:-----|:----|:----|:-----|
> > > | Enron | Batch | 3.27 ± 0.14     | 4.66 ± 0.42    | 4.69 ± 0.51    | 48.19 ± 0.58    | 97.30 ± 0.70     | 10.32 ± 0.10  | 17.11 ± 0.09  | 66.42 ± 0.36   |
> > > | Enron | Window | 3.97 ± 0.10     | 5.44 ± 0.21    | 5.44 ± 0.19    | 46.00 ± 0.97    | 86.02 ± 0.52     | 11.43 ± 0.11  | 17.12 ± 0.27  | 58.17 ± 0.66   |
> > > | UCI | Batch | 2.71 ± 0.23     | 3.48 ± 0.28    | 3.50 ± 0.33    | 28.63 ± 0.80    | 87.15 ± 0.57     | 5.73 ± 0.11   | 10.23 ± 0.33  | 32.93 ± 1.05   |
> > > | UCI | Window | 8.40 ± 0.24     | 10.07 ± 0.12   | 10.37 ± 0.17   | 38.10 ± 0.92    | 95.46 ± 0.62     | 15.63 ± 0.19  | 16.78 ± 0.38  | 37.86 ± 0.97   |
> > > | MOOC | Batch | 74.51 ± 0.83    | 82.40 ± 1.66   | 88.52 ± 2.52   | 293.57 ± 2.60   | 666.43 ± 38.50   | 136.00 ± 0.25 | 140.94 ± 0.48 | 340.66 ± 9.61  |
> > > | MOOC | Window | 72.15 ± 2.78    | 79.16 ± 4.34   | 83.47 ± 0.44   | 278.63 ± 5.28   | 613.75 ± 37.14   | 117.93 ± 0.48 | 127.60 ± 0.61 | 336.95 ± 0.89  |
> > > | Wiki. | Batch | 16.98 ± 0.87    | 19.73 ± 0.57   | 19.57 ± 0.87   | 75.19 ± 1.24    | 140.82 ± 1.33    | 18.80 ± 0.32  | 30.23 ± 0.41  | 85.28 ± 1.37   |
> > > | Wiki. | Window | 19.96 ± 0.50    | 22.46 ± 0.71   | 23.46 ± 1.00   | 78.16 ± 0.85    | 139.29 ± 0.64    | 21.43 ± 0.31  | 32.27 ± 0.73  | 85.56 ± 0.76   |
> > > | LastFM | Batch | 380.99 ± 12.12  | 370.06 ± 0.17  | 383.59 ± 23.13 | 1162.63 ± 7.73  | 3689.96 ± 81.35  | nan | nan | nan |
> > > | LastFM | Window | 294.37 ± 0.60   | 314.94 ± 0.56  | 315.01 ± 0.46  | 1045.44 ± 14.71 | 3381.94 ± 16.44  | nan | nan | nan |
> > > | Myket | Batch | 481.59 ± 7.94   | nan | 512.37 ± 20.87 | 962.83 ± 86.92  | 973.83 ± 4.48    | 626.97 ± 7.25 | 640.49 ± 7.10 | 993.96 ± 43.32 |
> > > | Myket | Window | 544.35 ± 14.55  | 559.54 ± 11.55 | 553.53 ± 5.20  | 880.56 ± 75.42  | 1067.29 ± 6.12   | 618.84 ± 6.33 | 628.36 ± 6.55 | 948.99 ± 5.81  |
> > > | US L. | Batch | 2.07 ± 0.10     | 2.57 ± 0.13    | 2.47 ± 0.15    | 23.66 ± 1.09    | 49.10 ± 0.84     | 6.88 ± 0.06   | 8.26 ± 0.13   | 34.72 ± 0.91   |
> > > | US L. | Window | 1.28 ± 0.13     | 1.64 ± 0.11    | 1.56 ± 0.10    | 13.20 ± 0.16    | 33.15 ± 0.16     | 3.26 ± 0.12   | 4.69 ± 0.11   | 24.21 ± 0.54   |
> > > | UN Tr. | Batch | 39.80 ± 0.28    | 46.56 ± 0.55   | 44.54 ± 0.15   | 769.15 ± 20.48  | 598.23 ± 0.26    | 136.72 ± 0.45 | 262.35 ± 0.59 | 361.83 ± 2.14  |
> > > | UN Tr. | Window | 2.28 ± 0.11     | 4.17 ± 0.12    | 4.41 ± 0.09    | nan | nan | 24.08 ± 0.22  | nan | nan |
> > > | Can. P.   | Batch | 2.65 ± 0.14     | 4.92 ± 0.27    | 4.94 ± 0.32    | 108.18 ± 0.41   | 191.27 ± 1.12    | 11.12 ± 0.16  | 29.76 ± 0.38  | 85.84 ± 0.83   |
> > > | Can. P.   | Window | 0.56 ± 0.12     | 2.30 ± 0.15    | 2.27 ± 0.13    | 75.20 ± 0.51    | nan | 4.68 ± 0.18   | nan | nan |
> > >
> > > We will include the runtime of all datasets and models in the camera-ready version.

---

> ### Comment · Reviewer_sDdG · 2024-12-03
> **Thanks for your further clarifications**
>
> I appreciate the authors for their further clarifications of (1) deeper insights regarding the flaws of the current evaluation protocol and (2) preliminary experiment results about runtime. I believe these results would largely increase the soundness of this research; therefore, I'm inclined to increase my contribution score from 1 to 2, my soundness score from 2 to 3, and an overall rating from 3 to 5.
>
> Please make sure to integrate our discussions during the rebuttal period into the camera-ready version.

---

### Official Review · Reviewer_Bi4c · 2024-11-03

**Soundness:** 3
**Presentation:** 3
**Contribution:** 2
**Rating:** 6
**Confidence:** 4

**Summary:**

This work introduces a reformulation of the dynamic link prediction (DLP) task as link forecasting (LF). LF differs from DLP in that LF ensures that each batch corresponds to a fixed time resolution. This small change to the problem statement yields relatively significant changes to the observed accuracies of these models on benchmark datasets.

**Strengths:**

1. The problem with variable-time batch construction for DLP is made quite clear in section 3. The examples are easy to understand
2. The experiments in section 4 are nearly exhaustive. The models cover the big TGNNs out there, and the datasets cover many of the major datasets out there.
3. The problem statement is clearly articulated and easy to understand. I had no problem implementing the discrete time version of the data loader in an afternoon of work. This clarity is commendable.

**Weaknesses:**

1. While the problem with DLP seems clear, it is hard for me to see why this requires the definition of a new task given that it's a straightforward modification of an existing one. I would recommend that the authors expand on this in the work to further draw the distinction, _or_ to assert that this is the way that DLP should be done in the future.
3. It appears that the models were not hyperparameter tuned for this new task. It stands to reason that because the new involves training on potentially quite different batch structures than the batches those hyperparameters were tuned for, that the reported model performance might be an underestimate. I would suggest that the authors investigate the hyperparameter sensitivity of the models training using LF.
4. The experimental details are a bit hard to follow in section 4. It appears that what was done is the authors trained a model using the horizon-based strategy, and then used this model to evaluate on both the horizon and batch based evaluation strategies. They then reported the performance differences. I would ask that the authors clarify these experimental details.

**Questions:**

1. It's unclear to me what the authors mean when they assert that DLP yields different amounts of information loss for different models. It seems to me that, if the batch size is fixed across all models, then it seems to me that the information loss (or really, the inter-batch leakage) will be equivalent. In what ways are they not? Or is the assertion that because batch size is often treated as a (often implicit) hyperparameter, so model comparisons can be implicitly unfair?
2. Does LF suffer from the same temporal training-leakage that DLP does when h is larger than infinitesimal?
3. Which is more significant? Batches containing variable time gaps, or the identified temporal leakage?
4. The primary motivation to use larger batch sizes is the reduction of wallclock time during training. How does the training time for LF scale as a function of h?
5. How do the results reported in table 2/3 change if the model was trained using a DLP pipeline and then evaluated on both DLP/LF?

---

> ### Author Response · Authors · 2024-11-21
>
> We appreciate your evaluation of our work and are pleased that you consider the clarity of our work "commendable".
> We provide clarifications to your comments and questions below:
>
> Commenting on your first point:
> Thank you for pointing this out, we are happy to clarify: The current formulation of dynamic link prediction allows tuning the batch size as a hidden hyperparameter. This may be for technical reasons, e.g. due to limited GPU memory, or for performance reasons where tuning the batch size to a given dataset and model improves model performance. However, if the batch size is tuned per model, this results most likely in different batch sizes for each model. Consequently, they consider potentially very different tasks, depending on how different the chosen batch sizes are. To address this, we propose to adopt "batching" in terms of fixed-size time windows, and we argue that those time windows should be chosen based on the specific dataset. Of course, this may be interpreted as modifying the existing dynamic link prediction task, however, by choosing a different name inspired by the time series analysis literature, we want to make the distinction clear: no fixed-size time windows vs. fixed-size time windows.
> We added further clarifications in the main text of the uploaded revision to emphasize that our evaluation method is designed to **replace** the batch-based evaluation.
>
> Regarding two and three:
> It appears there was a slight misunderstanding about our experimental details.
> We trained all models using only the batch-based approach.
> Afterward, we evaluated all models using both the batch-based and our proposed time-window-based approach and reported the performance differences.
> We improved the experimental details in the uploaded revision for more clarity.
> Since we did not change the training procedure that we took over from Yu et al. [1] and reused the best-performing model configurations that were found in their hyperparameter tuning, we argue that no additional hyperparameter tuning is necessary.
> ### Responses
> 1. We mean the second thing that you mention. As long as the batch size is kept the same for all models, no unfairness arises. However, the batching might still impose an artificial ordering on temporal edges with the same timestamp (information loss) or lead to parallel processing of edges with different timestamps and treating them as happening simultaneously (information leakage). An unfairness issue arises when the batch size is treated as an implicit hyperparameter that is tuned on a per-model basis. Then, different models may end up with different batch sizes, which changes the characteristics of the prediction problem, making it easier or harder because more or less information of the original edges' timestamps is retained. Consequently, models that use different batch sizes should not be directly compared because they address a different prediction task.
> 2. It depends on how we think about the prediction task. A long time horizon essentially coarse-grains the links' temporal resolution, which is like saying "all these links appear during the same time window". However, we explicitly incorporate this in the problem formulation, making this a deliberate choice based on the problem domain. In Appendix C, we suggest plausible time horizons for the considered datasets. We argue that the time horizons should be defined by the characteristics of the dataset or the required granularity for the considered problem. For example, it may not be required to predict whether a customer will purchase a certain product within the next second; making such a prediction for the next day or week may be sufficient.
> 3. Both the varying time gaps and the information leakage are significant. For continuous-time datasets with inhomogeneously distributed temporal activity (such as Enron or UCI), we observe substantial performance changes for all models when comparing both evaluation approaches because of the varying time gaps. For discrete-time datasets, we observe a substantial drop in performance using the time-window-based approach compared to the batch-based approach for memory-based models because of the identified information leakage.
> 4. This depends on the specific interaction patterns. However, generally, a larger $h$ will produce more links per time window, thereby it reduces the wallclock time of training due to better parallelization. Note that (as pointed out above) we, for now, only consider the evaluation and not the training. In this sense, the "training time" would only be impacted if you consider the evaluation on a validation set, e.g. for early stopping, as part of the training.
> 5. As pointed out above, using a DLP pipeline and then evaluating both DLP/LF is how we conducted our experiments. Using the time-window-based approach for training is left for future work.
> ### References
> [1] Le Yu et al. Towards Better Dynamic Graph Learning: New Architecture and Unified Library. In NeurIPS, 2023.

---

> > ### Comment · Reviewer_Bi4c · 2024-11-27
> >
> > I thank the authors for their responses. They've provided clear answers and with their answers, i feel that this is a valuable contribution to the temporal graph learning literature. I've increased my score accordingly.

---

### Official Review · Reviewer_P4Ga · 2024-11-03

**Soundness:** 3
**Presentation:** 2
**Contribution:** 2
**Rating:** 5
**Confidence:** 2

**Summary:**

In this paper, the authors considered the evaluation of dynamic link prediction on both discrete-time dynamic graphs (DTDGs) and continuous-time dynamic graphs (CTDGs). They first provided a series of empirical analysis results to demonstrate the limitations of existing batch-based evaluation strategies. A novel time-window-based approach was further proposed to address these limitations. The authors have also validated the effectiveness of the proposed evaluation approach on various public DTDGs and CTDGs datasets.

**Strengths:**

**S1**. The authors have conducted a great amount of experiments to illustrate the limitations of existing techniques as well as the effectiveness of the proposed method.

**S2**. The limitations of the evaluation of dynamic link prediction are significant but seldom considered and addressed in most existing studies.

**Weaknesses:**

**W1**. Some statements regarding the research gaps of existing techniques and motivations of this paper are weak, unclear, or confusing.

  According to my understanding, this study focuses on the evaluation of dynamic link prediction. However, the title of this paper uses the terminology 'temporal graph learning', which may not be consistent with the major topic of this study. From my perspective, learning may also include the training procedure (e.g., training algorithms, training losses, etc.), addition to evaluation of the inference procedure.

  The author claimed that 'within each batch, edges are typically treated as if they occurred simultaneously, thus discarding temporal information within a batch'. To some extent, I do not agree with this statement. According to my understanding, nodes or edges in most TGNNs are encoded as embeddings (i.e., low-dimensional vector representations), which usually involves the **temporal encodings**. In this sense, the temporal information has not been discarded. It is recommended to give some more toy examples about why and how is the temporal information discarded, especially for the case with temporal encodings.

  I also respectfully disagree with the definition of dynamic link prediction in Section 2, which was claimed to 'predict whether $(u, v) \in B_i^+$ or $B_i^-$ in terms of batches $B_i^+$ and $B_i^-$. According to my understanding, a dynamic link prediction model should be able to predict all the possible edges at a specific timestamp but not within a batch.

  The authors claimed that existing techniques may hinder the fair comparison of methods. However, it is unclear for me how to define and measure the fairness of comparison. A formal definition regarding this point is also recommended.

  From my perspective, using 'temporal link prediction' and 'temporal link forecasting' as two terminologies with different definitions may not be a good presentation, which may result in potential ambiguity issues, since they are more likely to be synonyms in natural languages. It is recommended to use a clearer terminology to replace 'dynamic link forecasting' (e.g., batch-based and window-based evaluation) that can help better distinguish between 'temporal link prediction' and 'temporal link forecasting'.

***

**W2**. There seems to be some flaws for the proposed evaluation approach.

  In Section 3.2, the authors discussed one possible limitation that the proposed approach 'cannot preclude memory overflows entirely'. A possible solution was then discussed, which 'splits large time windows into smaller batches for GPU-based gradient computation'. In this sense, the proposed method still used the old bath-based technique, which may still 'ignore the temporal information within each batch', as claimed at the very beginning of this paper.

  According to my understanding, the proposed method may also suffer from the empty window issue, where there are no edges in a 'pre-defined' window, due to the heterogeneous distribution of temporal edges. However, there are no discussions regarding this limitation and possible solutions.

***

**W3**. Some details of experiments need further clarification. Some additional experiments are also recommended.

  In the empirical analysis of this paper, NMI is a significant metric to 'measure the temporal information that is retained after dividing edges into batches'. However, the formal definition regarding how to compute such a measurement is not given. As a result, it is still hard for me to understand the physical meaning of NMI using in the empirical analysis.

  According to my understanding, the proposed time-window-based approach introduces another hyper-parameter $h$. However, there seems no parameter analysis regarding different settings of $h$ (like the empirical analysis shown in Fig. 3).


***

**W4**. Although the authors have provided a great amount of empirical results to demonstrate the limitations of existing techniques and validate the effectiveness of the proposed approach, it is better to provide some theoretical guarantees w.r.t. the evaluation of dynamic link prediction.

**Questions:**

See **W1**-**W4**.

---

> ### Author Response · Authors · 2024-11-21
>
> We are grateful for your constructive feedback and are happy to see that you find that the problems we address "are significant but seldom considered and addressed in most existing studies".
> We address your comments in the following.
> Note that due to space constraints, our response is split into multiple comments:
>
> ### W1
>
> Regarding your disagreement with our task definition:
> We agree with your understanding of the task that "a dynamic link prediction model should be able to predict all the possible edges at a specific timestamp but not within a batch".
> Indeed, as we state in l. 128-129, "Given time-stamped edges up to time t, the goal of dynamic link prediction is to predict whether an edge (v, u, t+1) exists at future time t+1".
> While this definition is commonly used in recent papers on dynamic link prediction, to the best of our knowledge, none of those works have actually adopted an evaluation procedure that would exactly match this task definition.
> For computational efficiency, in practice, edges are split into fixed-size batches.
> Therefore, simplifications are made for model training and evaluation, i.e. batches that may span many timestamps are used.
> The Edges in each batch can then be processed in parallel, but, as a result, negative edges are also sampled on a per-batch basis.
> Highlighting exactly this mismatch between the task definition and evaluation practices in recent literature is the key contribution of our work.
> Thus, in section 2 of our paper, we formulated the definition of link prediction using a prediction setting for edges per batch rather than per timestamp.
>
> Referring to your comment about temporal encodings:
> We are aware of the temporal encodings that most models employ as mentioned in lines 137-140.
> However, regardless of the model's abilities to encode the timestamp of individual edges, in the process of negative sampling, we nevertheless discard temporal information within batches, which leads to a biased evaluation:
> This is because - due to collision checks [1] - if an edge occurs as a positive sample in a batch, it cannot occur as a negative sample in the same batch even if it occurs with a different timestamp.
> Thus, while the prediction is made using the positive sample's timestamp, it is ignored for the remaining duration of the batch during evaluation, essentially discarding the information of when the prediction for the edge was made within the duration of the batch.
> We thank the reviewer for highlighting this possible misunderstanding.
> We have edited the text in the uploaded revision to clarify this point.
>
> Regarding your comment about fairness:
> Developing a formal definition of "fairness" for the comparison of dynamic link prediction methods is an interesting suggestion for future research. However, even in the lack of such a formal definition, in our work, we argue that fairness is not given under the current dynamic link prediction setup because the batch size is effectively a free hyperparameter that influences the difficulty of the prediction task. As we illustrate in Fig. 1, choosing different batch sizes can drastically affect the resulting time windows defined by batches. Moreover, we show empirically that real networks exhibit diverse interaction patterns (Fig. 2) and that tuning the batch size has a large effect on the resulting duration of batches (Fig. 3). Consequently, it is expected that model performance changes with the batch size, which makes it possible to tune the task to the chosen model instead of tuning the model to the task. To address this issue, we propose _dynamic link forecating_, which is based on fixing the time horizon to make model performance comparable across different models because the task is kept the same.
> We argue that this approach is, in fact, more fair than the practice currently adopted in the community.

---

> > ### Author Response · Authors · 2024-11-21
> >
> > ### W2
> >
> > Regarding your comment that splitting each time window further into batches leads to the same problems as a batch-based evaluation:
> > In a sense this is correct.
> > However, as we point out in lines 382-391, our approach does not "preclude information loss entirely", but instead "controls the information loss" ensuring that this loss of temporal information is consistent for all time windows even if the number of links in certain time windows exceeds the memory of the used GPU.
> > Crucially, we require that for all batches that correspond to the same time window not only the temporal information within each batch is discarded but also across all batches from this time window.
> > Thus, we deliberately discard information inside each time window which instead of preventing intra-time-window information loss, prevents information leakage.
> > Thank you for pointing this out, we clarified it further in the uploaded revision.
> >
> > Regarding your comment about time windows with no edges:
> > Yes, that is correct. It is possible that time windows are empty; in this case, the correct prediction would be that no interactions occur during that time window. Could you elaborate on why this would be an issue or why it would be expected that interactions should occur during each time window?
> >
> > ### W3
> >
> > Regarding your comment about NMI:
> > Normalized Mutual Information (NMI) is a common information-theoretic measure [2], which is often used in the context of clustering and community detection to compare two different clusterings.
> > It is based on mutual information, which for two random variables $X$ and $Y$ captures the bits of information that we gain about the outcome of $Y$ if we know the outcome of $X$ and vice-versa.
> > The specific value of mutual information depends on the entropy of the underlying random variables, and is thus difficult to compare across different settings.
> > To address this issue, normalized mutual information provides a measure between zero and one that is normalized based on the entropies of the underlying random variables.
> > In the context of our work, we use the NMI to capture the loss of information in batch-based evaluations, or - in other words - how much bits of information about the timestamps of edges we gain based on the batch number only.
> > We use the standard implementation in the Python library scikit-learn [3] to compute NMI.
> > We note that mutual information-based measures have frequently been used to evaluate information loss that is due to time aggregation in temporal graphs [4,5]. We have added a more detailed explanation in the form of an additional Appendix G to the uploaded revision.
> >
> > Regarding your comment about "hyper-parameter" $h$:
> > We argue that the horizon h should not be treated as a hyperparameter. Instead, it should be determined by the characteristics of the prediction task, see section 3.2. In Appendix C, we also suggest realistic values for h for the considered datasets. However, we added a plot (Figure 6 in the uploaded revision) that showcases the number of links per time window for various time horizons, similar to Fig. 3, in the appendix.
> >
> > ### W4
> >
> > May we ask what kind of properties the reviewer has in mind? It may be difficult, if not impossible to provide general guarantees that hold for all possible models and datasets because we are proposing an evaluation approach, not a specific model or metric. We are also not aware of any theoretical guarantees in the context of the dynamic link prediction task.
> >
> > ### References
> >
> > [1] Farimah Poursafaei, Shenyang Huang, Kellin Pelrine, and Reihaneh Rabbany. Towards Better Evaluation for Dynamic Link Prediction. In NeurIPS, 2022.
> >
> > [2] Vinh et al., Information Theoretic Measures for Clusterings Comparison: Variants, Properties, Normalization and Correction for Chance, J. Mach. Learn. Res. 11 (3/1/2010), 2837–2854.
> >
> > [3] https://scikit-learn.org/stable/modules/generated/sklearn.metrics.normalized_mutual_info_score.html
> >
> > [4] Pfitzner, R., Scholtes, I., Garas, A., Tessone, C. J. & Schweitzer, F. Betweenness Preference: Quantifying Correlations in the Topological Dynamics of Temporal Networks. Phys. Rev. Lett. 110, 198701 (2013).
> >
> > [5] Weng, T., Zhang, J., Small, M. et al. Memory and betweenness preference in temporal networks induced from time series. Sci Rep 7, 41951 (2017). https://doi.org/10.1038/srep41951

---

> ### Comment · Reviewer_P4Ga · 2024-11-26
>
> I appreciate the authors' responses and revisions, which address some of my concerns. However, some of my concerns remain, especially for the **interpretations about fairness**, which is also related to the **parameter setting of $h$**.
>
> In the response to W1, The authors argued that most existing evaluation strategies contain a hyper-parameter of batch size, whose value may significantly affect the evaluation results. It is a significant observation. Although the authors argued that $h$ in this paper is not a hyper-parameter (in their response to W3), I still respectfully disagree with it. As I can check in Appendix C of the revised paper, **$h$ seems to still be set based on some intuitive background knowledges** (e.g., the dynamics of Reddit are fast, so $h$ was set to 15min). Some of these intuitions are also unclear (e.g., why we set $h = 15min$ but not $h = 10min$ due to the fast dynamics of Reddit). In this sense, **$h$ still plays a role similar to hyper-parameter**. Thus, **we cannot highlight that the proposed method is more fair than existing techniques**, from my perspective.
>
> In my expectation, $h$ should be a deterministic unique value calculated by an equation regarding characteristics of the prediction task. In this sense, **formal definitions and theoretical guarantees are**, to some extent, **necessary** in making such a contributions. However, as claimed by the authors, they could not give related definitions and theoretical analysis.
>
> My another concern is about the definition of NMI. I know that NMI is a widely-used metric but I believe that different tasks may have different definitions regarding this metric. For instance, NMI of community detection should be different from that in this paper. People familiar with community detection may not be familiar with the evaluation of temporal link prediction. It is recommended to give the formal definition of NMI using the notations in this paper (e.g., &B_i^+& and &B_i^-&) rather than just using the high-level notations of X and Y in Appendix G of the revised paper.
>
> Therefore, I keep my score.

---

### Official Review · Reviewer_m8vp · 2024-11-03

**Soundness:** 3
**Presentation:** 3
**Contribution:** 2
**Rating:** 6
**Confidence:** 3

**Summary:**

This paper highlights an overlooked issue in the evaluation of dynamic link prediction task: fixed batch-size evaluation alters the task properties, as for continuous-time temporal graphs, leading to inconsistent duration evaluations across batches; for discrete-time temporal graphs, leading to possible data leakage due to additional introduced temporal dependencies

To explore this issue in depth, the paper first defines a quantitative metric, NMI, to measure information loss, and conducts extensive empirical analysis to demonstrate how fixed batch sizes distort the task setting. It then formulates a fairer setting, *link forecasting*, enabling consistent time durations for each evaluation batch. Finally, the authors reproduce experiments on existing methods within this reformulated task to reveal their true performance.

**Strengths:**

*S1* This paper addresses an important yet overlooked issue in temporal link prediction task, i.e., fixed batch size evaluation can distort the task itself by losing or introducing extra information.

*S2* The authors provide extensive data illustrations and quantitative results to facilitate understanding, demonstrating that each dataset has a distinct interaction distribution and how fixed batch-size evaluation can alter the task characteristics.

*S3* This paper formulates a new task setting, *link forecasting*, and offers implementation and reproduction of existing methods to provide valuable insights.

**Weaknesses:**

I appreciate the issue raised in this paper and the extensive empirical analysis that clarify the motivation behind the study. However, for the experiments on existing methods within the formulated link forecasting task, I think further discussion is required, e.g., **the reasons for performance changes across diverse settings can be addressed.**

 The authors explain why memory-based methods tend to experience performance degradation on discrete-time graphs, which is appreciated. However, it would be beneficial to discuss why other methods might improve in this setting. Additionally, the performance trends for continuous-time graphs appear mixed, potentially due to specific dataset characteristics. I think a more in-depth discussion on these points would enhance the paper.

**Questions:**

Please refer to the "Weaknesses" section.

---

> ### Author Response · Authors · 2024-11-21
>
> We appreciate the valuable feedback and are glad that you think our paper addresses an "important yet overlooked issue" and provides "valuable insights".
> We agree that the explanation for the increase in performance for most non-memory-based models on most discrete-time datasets is incomplete and are happy to provide further information in the uploaded revision of our paper as well as in the following:
>
> We observe an increase in performance for our time-window-based approach compared to the batch-based approach for most non-memory-based models on most discrete-time datasets because for $h = 1$ our time-window-based approach uses all edges from one snapshot for each time window preventing overlapping batches that contain edges from multiple snapshots.
> In contrast, the batch-based approach creates batches that "stretch across snapshots and discard the temporal ordering of edges from different batches" (lines 311-312).
> When making predictions, only edges that occur _before_ the current batch may be used.
> However, since batches stretching across snapshots contain edges from multiple snapshots, all edges belonging to those snapshots must not be used to make predictions -- because they have not occurred _before_ the batch.
> Therefore, in the case of overlapping batches, not all available information can be used to make predictions.
> However, with our proposed link forecasting task definition, this is not an issue, resulting in the observed performance increase.
>
> Thank you for suggesting a more in-depth discussion on the performance trends for continuous-time graphs.
> In Appendix C, we include a more in-depth discussion on model performance for link forecasting using realistic time horizons on continuous-time graphs.

---

> > ### Comment · Reviewer_m8vp · 2024-11-28
> >
> > Thank you for the authors' response. I appreciate the proposed setting and extensive empirical results presented in this paper. As a result, I have increased my score to 6.

---

### Comment · Area_Chair_rK1f · 2024-11-23

Dear Reviewers,

The authors have uploaded their rebuttal. Please take this opportunity to discuss any concerns you may have with the authors.

AC

---

### Author Response · Authors · 2024-12-04

We thank all reviewers for the constructive discussions and their helpful suggestions. We are happy to see that you think that the "overall presentation is clear" with "clarity [that] is commendable" and unanimously agree that the soundness of our paper is good. We further appreciate that the reviewers agree that our paper "addresses an important yet overlooked issue" and contains "a great amount of experiments to illustrate the limitations of existing techniques as well as the effectiveness of the proposed method".

Furthermore, we are glad that we were able to **clear up most concerns and, especially, all concerns of the reviewers m8vp, Bi4c, and sDdG**. Specifically, we improved our explanations for the performance changes due to the comments made by reviewer m8vp. Thanks to suggestions made by reviewer sDdG, we were able to provide additional empirical evidence that support our explanations of the performance changes as well as our claims about the prediction task being skewed and unfair. Thanks to reviewer sDdG, a runtime evaluation showing comparable results for our method and the batch-based approach supporting our claims about computational cost in our manuscript will be added in the camera-ready version. We were also able to correct typos and improve the general formulations in our manuscript to avoid potential misunderstandings thanks to the comments of the reviewers. Following the suggestion made by reviewers P4Ga and sDdG, we added an additional appendix that explains the NMI measure in more detail and agree with Reviewer sDdG who thinks "it is pretty clear now". However, we are happy to follow reviewer P4Ga's suggestion and will provide examples that use our notations that will help with the understanding in the camera-ready version.

Next, we provide further clarifications about the unresolved concerns of reviewer P4Ga about the time horizon $h$:

We agree that the horizon $h$ is a parameter that can be changed. We further agree that there might not be a distinct value that makes sense for all forecasting tasks (e.g. forecasting links on Reddit, as mentioned by reviewer P4Ga). However, in contrast to the batch size - a **hyperparameter that is typically tuned for best performance** - the **horizon $h$ is a parameter that defines the task and cannot be tuned**.

Additionally, while it might make sense to calculate a deterministic unique value for some tasks based on the dataset's characteristics as reviewer P4ga proposes, there are also many cases where the **time horizon of the task is very different from the optimal time horizon based on the datasets characteristics**. A temporal network of train connections, for example, might have an optimal horizon $h=24h$ since most trains follow a daily schedule. Instead of learning the fixed schedule, the task for the forecasting model could be to learn unforeseen deviations from the schedule to react accordingly, e.g. by rerouting trains. As such, the forecasting horizon $h$ should be based on the minimum reaction time that is needed to make, e.g. the rerouting of a train possible, but at the same time as short as possible to give the model access to the most up-to-date information, e.g. $h=60min$ instead of $24h$ as suggested by the data.

Thus, while the separate problem of **finding an optimal time horizon with theoretical guarantees** is related and can help for some link forecasting tasks where no specific time horizon is required, it is **out of scope for our work**. For further insights into this separate problem, we kindly refer to papers addressing this separate problem:

Clauset, A. & Eagle, N. Persistence and periodicity in a dynamic proximity network. in Proceedings of the DIMACS Workshop on Computational Methods for Dynamic Interaction Networks (2007).

Darst, R. K. et al. Detection of timescales in evolving complex systems. Sci Rep 6, 39713 (2016).

Petrović, L. V. et al. Higher-Order Patterns Reveal Causal Timescales of Complex Systems. Preprint at https://doi.org/10.48550/arXiv.2301.11623 (2023).

In general, our discussion with reviewer P4Ga about the task definition of dynamic link prediction and the time horizon $h$ highlights the complexity of the problem itself. Therefore, we believe that by pointing out problems in the current evaluation approaches and by suggesting a replacement that fixes these problems, our work is a valuable contribution that helps to bring temporal graph learning closer to making real-world impact.

---

### Meta-Review · Area_Chair_rK1f · 2024-12-20

**Metareview:**

This paper addresses the limitations of existing evaluation strategies for dynamic link prediction (DLP) tasks in temporal graphs, proposing a new task setting called link forecasting (LF). The authors highlight how fixed batch-size evaluation distorts task properties and can lead to issues like temporal dependencies or information leakage. They introduce a novel time-window-based evaluation approach, backed by extensive empirical results, to provide more accurate performance assessments of existing models.

While the proposed model shows promising results, there are several weaknesses that still need to be addressed:

1. The batch-oriented evaluation problem in continuous-time graphs only affects memory-based methods. Recent sequence-based models (e.g., GraphMixer, DyGFormer) ensure that each node is assigned all its historical neighbors immediately before the prediction time, thereby avoiding the issues described.
2. The proposed LF setting doesn’t fully resolve the problem, as it still faces information leakage when the edges in a time window exceed GPU memory. While the authors suggest that discarding information within each time window can mitigate information leakage, this can also be achieved when using batch-based evaluation.
3. The time-window approach introduces an unfixed number of edges per batch, which could impact training stability and convergence, but this is not discussed in the paper.

Based on these weaknesses, we recommend rejecting this paper. We hope this feedback helps the authors improve their paper.

**Additional Comments On Reviewer Discussion:**

In their rebuttal, the authors made several improvements, including clarifications and updates to the presentation, which help reviewers better understand the contributions of the paper. However, the reviewers’ concerns regarding the novelty of the paper and the effectiveness of the proposed method, remain unresolved. As a result, I recommend rejection based on the reviewers’ feedback.

---

### Decision · Program_Chairs · 2025-01-22

Reject